# The Mother–Child Dyad Adipokine Pattern: A Review of Current Knowledge

**DOI:** 10.3390/nu15184059

**Published:** 2023-09-19

**Authors:** Jolanta Lis-Kuberka, Małgorzata Pupek, Magdalena Orczyk-Pawiłowicz

**Affiliations:** Department of Biochemistry and Immunochemistry, Division of Chemistry and Immunochemistry, Wroclaw Medical University, M. Skłodowskiej-Curie 48/50, 50-369 Wroclaw, Poland

**Keywords:** mother–child dyad, adipokines, maternal and cord plasma, newborn/infant plasma, human milk, gestational diabetes mellitus, obesity, lifestyle diseases

## Abstract

An important role in the network of interconnections between the mother and child is played by adipokines, which are adipose tissue hormones engaged in the regulation of metabolism. Alternations of maternal adipokines translate to the worsening of maternal insulin resistance as well as metabolic stress, altered placenta functions, and fetal development, which finally contribute to long-term metabolic unfavorable conditions. This paper is the first to summarize the current state of knowledge concerning the concentrations of individual adipokines in different biological fluids of maternal and cord plasma, newborn/infant plasma, milk, and the placenta, where it highlights the impact of adverse perinatal risk factors, including gestational diabetes mellitus, preeclampsia, intrauterine growth restriction, preterm delivery, and maternal obesity on the adipokine patterns in maternal–infant dyads. The importance of adipokine measurement and relationships in biological fluids during pregnancy and lactation is crucial for public health in the area of prevention of most diet-related metabolic diseases. The review highlights the huge knowledge gap in the field of hormones participating in the energy homeostasis and metabolic pathways during perinatal and postnatal periods in the mother–child dyad. An in-depth characterization is needed to confirm if the adverse outcomes of early developmental programming might be modulated via maternal lifestyle intervention.

## 1. Introduction

The relationship between the mother and child begins with fertilization and proceeds through the three trimesters of pregnancy, as supervised by a dedicated set of hormonal and immunological factors [1,2]. Pregnancy, as a physiological process, is the most nutritionally sensitive stage during a woman’s life, and, for this reason, the balanced state of a pregnant woman in the metabolic context translates into a lower risk of chronic lifestyle diseases developing in their offspring [3,4,5]. The first 1000 days of a child’s life (including prenatal and postnatal development) translate to the wellbeing and susceptibility to metabolic disorders in their later life. In particular, the maternal diet and physical activity before and during pregnancy have an impact on the fetus’s organ development, endocrine programming, and finally, the epigenetic programming of gene expression [6,7,8].

After delivery, the close nutritional and immunological contact between the mother and newborn is continued by breastfeeding, and therefore, the first twelve weeks, often called the fourth trimester, are as important for a mother’s and infant’s health as the three trimesters of pregnancy [9]. However, as for the prenatal period, during the perinatal and postnatal periods, the mother and her child must be treated as an inseparable dyad [10,11,12,13,14,15]. The mother–infant dyad constitutes a unique system of multi-level interaction, including affective, behavioral, and physiological functionalities, in which both parties directly affect each other [16,17,18] due to the complex network of biochemical and emotional signal transmission [19].

Among the main factors affecting the mother–child dyad are genetic factors, medical comorbidities (hyperglycemic state and obesity), psychiatric disorders, civilization disease, and difficult socio-economic conditions [19,20,21,22,23,24,25,26]. An unfavorable prenatal environment accompanying the developing fetus may disturb its harmonious development as well as metabolism and triggers mechanisms to ensure the best possible adaptation to changed conditions [27,28,29]. 

Pregnancy is related to hormonal changes that affect the accumulation of adipose tissue (about 30% of gestational weight gain) and the secretion of adipokines. This phenomenon is responsible for the alteration of maternal metabolism and significantly reduces the sensitivity of a pregnant woman to insulin [30]. During pregnancy, maternal insulin resistance increases and might translate to the increased transport of glucose and amino acids and fatty acids across the placenta [27]. The development and maturation of the placenta is disturbed as a result of maternal-impaired glucose metabolism, and it leads to vascular dysfunction, including increased angiogenesis, villous fibrinoid necrosis [31,32,33,34], and changes associated with inflammation and oxidative stress that can lead to chronic fetal hypoxia [35]. Finally, maternal hyperglycemia might induce an increase in fetal insulin resistance and translate into its disturbed growth [28,36,37,38] as well as a long-term health risk for infants [27]. 

The importance of nutrition in pregnancy and the lactation period and its lifelong consequences are well established. In fact, both excessive and insufficient weight gain during pregnancy are associated with poor perinatal outcomes for both the mother and child [5]. Barker’s hypothesis [39,40,41,42,43] links the adverse nutrition of children in the early stage of life (prenatal period) with a higher risk of the development of cardiovascular, metabolic, and endocrine diseases (e.g., obesity, diabetes, insulin resistance, hypertension, hyperlipidemia, lipotoxicity, cardiac hypertrophy, and finally, coronary heart disease and stroke) in later life [8,44,45,46]. The developing fetus’ organs undergo “fetal programming” during the prenatal period, which can be modulated via internal and external factors and finally translate to the postnatal period [29,44,47,48,49,50,51,52]. The intrauterine environment disturbed by maternal hyperglycemia and the molecular mediators involved in the regulation of appetite, metabolism, and energy balance, such as adiponectin, can disturb the infant’s metabolic status, leading to increased weight gain [53,54]. In recent years, the incidence of diabetes, including gestational diabetes mellitus (GDM), has increased significantly, and the percentage of overweight and obese mothers and children is alarmingly high [55,56,57,58,59,60]. 

Evidence from animal models clearly indicates adipose tissue and its metabolism as factors that alter development and maturation, with consequences into adulthood [61,62,63]. Adipose tissue, as an endocrine organ, has the ability to secrete a wide range of peptides, peripherally as well as neuroendocrinologically, which are defined as adipokines and include adiponectin, leptin, ghrelin, resistin, vaspin, visfatin, chemerin, and apelin, among others. Moreover, it acts as an important buffer system for the energy balance. The latest classification determines adipokines as inflammation-related factors which take part in energy-state homeostasis as well as in regulatory events of metabolism and fat tissue [64,65,66]. An altered adipokine profile is associated with many metabolic disorders such as obesity [67,68,69,70], diabetes [71,72], insulin resistance [73], and cardiometabolic disease [74,75].

Apart from adipose tissue synthesis, adipokines such as leptin, resistin, and visfatin are also synthesized locally by the placenta and released to the fetal and maternal circulation; however, their precise roles are still unknown [76,77,78,79]. Placenta-related adipokines might be involved in the regulation of maternal metabolism for both physiological and pathological pregnancy [78]. Breast milk adipokines may have a dual origin; namely, adipokines released by adipose tissue into the maternal blood due to the diffusion and/or a specific receptor can access the mammary gland [80,81] and may be produced locally by lactocytes [82,83,84]. The presence of adipokines in the placenta, the fetus, and later, in maternal milk suggests that these hormones play a pivotal role in prenatal and postnatal development [81,84,85,86]. As critical signaling peptides, they take part in the modulation of nutrient transport via the placenta, which directly translates into the growth and development of the fetus during pregnancy [86]. Additionally, adipokines are the modulators of the immunological system, and any alterations of their profile should be considered as risk factors for chronic low-grade inflammation associated with diabetes and obesity [81,87]. Moreover, the mild pregnancy-associated pro-inflammatory condition is further exacerbated in obese mothers [88,89,90].

The natural way of feeding plays a significant role in protecting against excessive weight gain and reduces the risk of the development of type 1 diabetes [91]. It is indicated as one of the key elements of the primary prevention of childhood overweight, obesity, and metabolic alteration [92,93]. According to the World Health Organization (WHO) [94], exclusive breastfeeding is recommended up to 6 months and, as a gold standard, is the best documented nutritional intervention so far, having an undeniably positive impact on the development of an infant in its later life [95,96,97,98,99,100,101,102,103,104]. However, maternal metabolic disorders, including obesity, translate to alterations of the immunological quantity and quality of breast milk [105,106,107]. 

Adipokines are suggested to act as a key element of a nutritional connection important for metabolic health affecting the mother–child relationship [108,109,110]. The number of articles evaluating adipokine profiles in one type of biological material increases each year [92,111,112]. However, the existing evidence, especially in maternal–infant dyads concerning adipokines and their association with the early development of offspring, remains less clear, with the published results being fragmentary and inconsistent. In light of the above, there is a need for the simultaneous characterization of adipokine profiles in the complex view, which is undoubtedly the maternal–infant dyad system, and to summarize the adipokine profiles for the biological samples of both maternal and newborn origins. The review includes the evaluation of the profile of adipokines in maternal–infant dyads in at least two biological fluids of different origin, namely maternal and cord plasma, neonate/infant plasma, milk, and placental tissues and its relations with perinatal risk factors.

## 2. Scope and Methodology

The search strategy was developed for MEDLINE (via PubMed) and then adapted to the Scopus database. The specific time frame of article searching was from 2000 to 10th March 2023. Only full-text human studies were included, with no date range restrictions. Conference papers, editorials, letters, commentaries, short surveys, and notes were excluded. The search strategy used the following terms, including Mesh terms: (adiponectin OR leptin OR resistin OR irisin OR ghrelin OR nesfatin-1 OR vaspin OR visfatin OR chemerin OR apelin OR adropin OR copeptin OR omentin OR dermcidin) AND human AND (serum OR plasma) AND/OR cord AND/OR (milk OR placental). The main criterion for including articles in this study was the determination of adipokine concentration in at least two different biological fluids of maternal and/or neonatal origin, namely maternal plasma and milk as well as cord and neonatal plasma, and additionally, placental tissue. Based on the search strategy and abstract evaluation, 117 articles were identified; moreover, additional records (*n* = 26) were revealed through the review of the reference lists of other studies. After removing duplicates (*n* = 35) and an article in a language other than English (*n* = 1), 107 papers were screened for eligibility. Articles which concerned the assessment of the adipokine pattern in only one biological material were excluded (*n* = 60). Finally, 47 studies were included to evaluate current knowledge on the adipokine patterns in the maternal–infant dyad (Figure 1). 

## 3. Adipokines in Maternal–Infant Dyad

In most of the selected studies (32 out of 47; 68.1%), the concentration of a single adipokine was evaluated in at least two different biological fluids (Table 1). The set of two different peptide hormones in the biological samples from a maternal–infant dyad were analyzed in 9 out of 47 studies (19.2%), three in 5 out of 47 (10.6%), and four in 1 out of 47 (2.1%) scientific reports. The state of knowledge in the field of the adipokine level analyzed parallelly in at least two different biological fluids of maternal and neonatal origin, namely maternal plasma and milk as well as cord and neonatal plasma, and additionally, placental tissue, are summarized in Table 2. Thirty-eight out of forty-seven analyzed reports show the data on adipokine levels for two different biological samples (maternal plasma and/or cord plasma or neonate’s plasma or milk or placental samples) (Table 1 and Table 2) and nine for a maternal–infant triad (maternal and infant plasma/serum and human milk or placental samples) (Table 2). Moreover, the reported relationships among the analyzed parameters, including perinatal risk factors, in mother–infant dyads are also summarized (Table 2).

Only seven reports presented data for three different biological materials, namely for maternal plasma, cord plasma, and placenta [113,127,128,143,146,148,156], and two papers analyzed three different biological materials in another layout, namely for maternal plasma, cord plasma and maternal milk, or infant’s plasma [119,144]. However, the most frequently studied pair of biological materials in the analyzed set of studies was that of maternal plasma and cord serum (23 out of 47), followed by maternal plasma and breast milk (8 out of 47) (Table 2). Three of the studies evaluated the adipokine levels in maternal plasma/serum and the placental samples [77,124,142]. Finally, two of the studies analyzed the concentrations of adipokines simultaneously in cord plasma and neonatal serum [114,138], and also two articles were focused on hormone patterns in neonatal serum/plasma and human milk [120,129].

Thirty-four out of 47 scientific papers parallelly analyzed the results for the biological samples of maternal and newborn origin. The remaining papers focused on two different biological materials, although both are of maternal origin (11 out of 47) or only neonatal origin (2 out of 47). 

### 3.1. Adiponectin 

A detailed analysis concerning the concentration of adiponectin in the maternal—infant dyad during the perinatal period is still missing. So far, eight reports have presented the data on the adiponectin levels for at least two different biological samples [28,77,114,115,116,117], but two of them are focused on GDM (Table 3) [113,118].

It was reported [118] that the concentration of adiponectin in the plasma of lean mothers was not significantly higher (5.0 ± 1.0 µg/mL) in comparison to obese mothers without and with GDM (4.1 ± 1.8 µg/mL and 3.8 ± 1.4 µg/mL, respectively), while in cord blood plasma, the adiponectin level was not related to any accompanying disorders (lean group: 20.7 ± 6.2 µg/mL; obese group without GDM: 20.8 ± 8.6 µg/mL and with GDM: 22.0 ± 7.3 µg/mL).

In 2017, Moran and coworkers [116] analyzed the effect of an antenatal lifestyle intervention in overweight and obese women on the adiponectin level in maternal and cord plasma. Their findings revealed that the change in dietary and lifestyle habits (combination of dietary and physical activity) by overweight or obese pregnant women did not translate into significant differences of adiponectin levels in women’s plasma and cord blood (7.8 µg/mL and 22.4 µg/mL, respectively) in comparison to standard care mothers (8.0 µg/mL and 22.3 µg/mL, respectively). Similar trends, but with slightly higher values, were presented by Rio-Age and coworkers [117], who showed that in maternal plasma that was collected at birth after the 37th week of pregnancy and complicated by infection, the adiponectin level was 13.5 ± 0.8 µg/mL, while in cord plasma, the level was 29.9 ± 3.1 µg/mL. 

In 2018, Shang’s group [113] analyzed oxidative stress in GDM mothers and their newborns and found that the level of adiponectin in maternal plasma (GDM: ~0.5 µg/mL and non-GDM: ~0.8 µg/mL) was approximately four times lower than in cord plasma (GDM: ~2.7 µg/mL and non-GDM: ~2.6 µg/mL), but adiponectin expression in the placental tissue was not detected. For the GDM cohort, maternal plasma adiponectin was lower than in the non-GDM group, while for cord plasma, it was higher in women with GDM in comparison to the non-GDM cohort. Additionally, the authors reported finding a positive correlation of adiponectin levels with markers of oxidative stress and with the quantitative insulin sensitivity check index. Shang and coworkers [113] also analyzed the relationship between maternal and cord adiponectin and macrosomia and reported that maternal plasma adiponectin was significantly lower for mothers who delivered a macrosomic baby (1.8 ± 0.6 µg/mL) in comparison to normal birth weight newborns (2.2 ± 0.9 µg/mL), while for cord plasma, they did not observe any significant differences between the analyzed cohorts (2.7 ± 0.5 and 2.7 ± 0.6 µg/mL, respectively).

A subsequent study showed completely different values, in terms of concentration levels, to those previously published. In 2016, Aydin and coworkers [115] analyzed adiponectin levels in intrauterine growth restriction (IUGR) neonates and their mothers and found that maternal serum adiponectin levels were similar for both analyzed groups (for IUGR: 4.0 ± 1.6 ng/mL and for normal: 3.7 ± 1.1 ng/mL, respectively), but for IUGR, the cord adiponectin level was significantly lower (3.3 ± 1.3 ng/mL) in comparison to the normal group (5.2 ± 3.1 ng/mL). Nevertheless, no correlation between the maternal and cord adiponectin level and neonatal birth weight was reported [115]. 

Meyer and coworkers [114] focused on searching for relationships between cord blood adiponectin and the risk of obesity development of up to 5 years of age. The cord blood high molecular weight (HMW) adiponectin (14.9 µg/mL) was higher than in 3-year-old children’s plasma (9.2 µg/mL) and showed positive trends with the newborn’s weight, fat, and lean body mass, and additionally, with BMI. Similarly, the adiponectin concentration of 3-year-old children is not related to the distribution of adipose tissue in children up to 5 years of age [114].

The simultaneous determination of adiponectin levels in two different maternal biological samples is limited to two studies only, namely plasma and milk [28] and plasma and placenta [77]. Mohamad’s group [28] found that the level of adiponectin in human milk, regardless of the lactation stage, was much lower (colostrum: 17.1 ± 8.8 ng/mL; mature milk from second month: 11.5 ± 8.5 ng/mL) than in maternal serum (8.6 ± 6.5 µg/mL and 7.6 ± 3.9 µg/mL in the second and third trimester of pregnancy, respectively). However, in both cases, a significant negative relationship with the development of the adiposity of infants up to the first year was detected. Haugen and coworkers [77] analyzed the adiponectin status in maternal samples in pre-eclamptic mothers and observed that the concentration of adiponectin in the plasma of the mothers with preeclampsia (18.3 ± 2.2 µg/mL) was significantly higher in comparison to physiological pregnancy (12.2 ± 1.1 µg/mL). On the other hand, the presence of adiponectin (based on the mRNA determination) in placental tissue was not detected for both groups.

### 3.2. Leptin

The knowledge concerning the leptin level in maternal–infant dyads, similar to adiponectin, is fragmentary. So far, only nine reports have presented data on the leptin levels for two different biological samples (maternal and cord plasma) [77,115,116,117,118,120,121,122,123], and two simultaneously characterize the leptin status in three different biological fluids: maternal plasma, cord plasma, and milk (Table 3) [113,119]. 

As was pointed out by Uebel and coworkers [118], the leptin levels in maternal plasma with a normal BMI was 9.3 ng/mL, but maternal obesity was related to a significant increase, namely for obese pregnant women with and without GDM levels were similar at 42.2 ng/mL and 46.0 ng/mL, respectively. On the other hand, the leptin concentration in cord plasma was slightly higher in both obese cohorts regardless of GDM occurrence (with GDM: 6.8 ng/mL and without GDM: 7.6 ng/mL, respectively) in comparison to the cord plasma level of the control cohort (5.9 ng/mL) [118].

In 2016, Aydin and coworkers [115] investigated leptin levels in maternal and cord serum of IUGR neonates and their mothers and concluded that the maternal leptin levels for the IUGR cohort (16.00 ± 9.80 ng/mL) and the normal group (19.7 ± 16.3 ng/mL) were comparable. Similarly, cord serum leptin levels in the IUGR group (18.0 ± 13.1 ng/mL) and in the control group (15.0 ± 8.0 ng/mL) did not differ significantly due to the large spread within the analyzed cohorts. Additionally, no correlation between the leptin level in the maternal and cord samples and neonatal birth weight was found [115]. 

Moran’s group [116] analyzed the impact of lifestyle intervention on the leptin level in maternal and cord plasma and found that the leptin levels did not differ significantly among the analyzed cohorts of overweight and obese pregnant women included in the intervention study aiming at changes in lifestyle and standard care for pregnant women for both maternal (54.2 ng/mL and 54.5 ng/mL, respectively) and cord plasma (13.1 ng/mL and 13.1 ng/mL, respectively) [116]. 

In 2018, Shang’s group [113] analyzed the leptin levels in women with GDM and their newborns in relation to oxidative stress and noted that the level of leptin in maternal blood (GDM: ~0.6. ng/mL and non-GDM: ~0.5 ng/mL) was lower than in cord plasma (GDM: ~0.7 ng/mL and non-GDM: ~0.6 ng/mL). The leptin level in placenta samples of the GDM group was ~0.5 ng/mg placenta, and for the non-GDM group, ~0.4 ng/mg placenta. Moreover, the leptin levels in maternal, cord plasma, and placenta were significantly higher in the GDM cohorts in comparison to the non-GDM group. In the same study, Shang and coworkers [113] presented the effects of leptin in maternal plasma, cord plasma, and placenta on the incidence of macrosomia. The level of placental leptin of mothers who delivered a macrosomic baby was significantly higher (16.9 ± 3.0 ng/mg) than for women who delivered normal weight newborns (12.6 ± 3.9 ng/mg). Similar results were obtained for the leptin level in cord plasma from a macrosomic cohort; namely, it was significantly higher (22.2 ± 3.5 ng/mL) than for the normal birth weight group (17.8 ± 3.6 ng/mL). In contrast, maternal plasma leptin was at a comparable level regardless of birth weight (group of mothers who bore a macrosomic baby: 17.4 ± 4.3 ng/mL and group of mothers who bore normal birth weight newborns: 16.7 ± 3.0 ng/mL) [113].

It was reported [121] that for term-born neonates with excessive gestational weight gain (EGWG), the leptin levels were higher than for the control group, significantly for cord serum (EGWG: 11.0 ng/mL and control cohort: 7.5 ng/mL), and not significantly for maternal serum (EGWG: 14.9 ng/mL and control cohort: 10.4 ng/mL)

Rio-Age and coworkers [117], apart from the cytokine and immunoglobulin profiles of antibiotic-treated pregnant women beyond the 37th week of gestation, analyzed the leptin level in maternal (22.6 ± 4.2 ng/mL) and cord plasma (8.1 ± 1.1 ng/mL). However the results differed from those previously published, probably due to the impact of pathological conditions associated with the course of pregnancy [117].

More detailed analysis in this area was recently reported by Garofoli’s group [119], who focused on the leptin status in maternal, cord plasma, and milk samples in relation to the gestational age and weight at birth. The maternal plasma leptin levels of mothers delivered preterm (75.6 ng/mL) and intrauterine growth-restricted (IUGR) newborns (71.8 ng/mL) were significantly higher than for mothers who delivered at term (44.5 ng/mL). In contrast, a reverse pattern was observed for cord serum, namely leptin contractions in cord sera in both groups (for preterm: 3.96 ng/mL and intrauterine growth-restricted 1.6 ng/mL newborns, respectively) were significantly lower than for the cord serum of newborns delivered at term (19.3 ng/mL). In the milk samples, the concentration of leptin was about one hundred times lower than in maternal plasma and was at an almost unchanged level regardless of the pathophysiological condition, namely 621 pg/mL for term, 622 pg/mL for preterm, and 844 pg/mL for IUGR newborns [119].

Pekal and coworkers [122] analyzed the leptin levels in maternal and cord serum depending on birth weight and their association with the newborn’s anthropometric measurements. Pekal’s group identified significant differences in the cord serum leptin levels among the term-born small for gestational age (SGA) (3.5 ng/mL), appropriate for gestational age (AGA) (6.3 ng/mL), and large for gestational age (LGA) (9.8 ng/mL) groups. On the other hand, no significant differences for the leptin level in maternal plasma regardless of birth weight (SGA: 19.2 ± 8.4 ng/mL; AGA: 16.7 ± 9.8 ng/mL; and LGA: 22.0 ± 11.9 ng/mL, respectively) were observed [122]. However, the associations between the cord serum leptin level, placental weight, and anthropometric data such as newborns’ head circumference, length, and weight at birth were found [122]. The latest study by Gök’s group [123] showed that the maternal and cord leptin levels were significantly higher for pre-eclamptic mothers (21.7 ng/mL and 11.5 ng/mL, respectively) in comparison to the physiological pregnancy (4.4 ng/mL and 3.3 ng/mL, respectively), and additionally, a positive correlation between the maternal and cord serum leptin levels and maternal BMI for both cohorts was identified [123].

The simultaneous determination of the leptin levels in two different biological samples (other than maternal and cord plasma) is limited to two studies only, namely maternal plasma and placenta [77] and newborn plasma and maternal milk [120]. Haugen and coworkers [77] analyzed the leptin status of pre-eclamptic mothers and found that the concentration of leptin in the plasma of pregnant women with preeclampsia was significantly higher than in the plasma of women in physiological pregnancy (34.4 ± 3.2 ng/mL and 22.3 ± 1.1 ng/mL, respectively). Simultaneously, the presence of leptin in placental tissue was confirmed and the level of leptin for the pre-eclamptic cohort was higher than for physiological pregnancy [77].

Chatmethakul and coworkers [120] compared the level of leptin in the plasma of children depending on the type of milk they were fed, namely their own mother or donor milk, and found that the leptin level in standard donor milk was significantly lower (3.8 pg/mL) than in maternal milk during the first or third week of lactation (580 pg/mL and 577 pg/mL, respectively) [120]. Additionally, their findings revealed that the leptin level in extremely premature infants’ plasma positively correlated (r = 0.4) with the concentration of the milk’s leptin regardless of the milk type. Moreover, in the plasma of preterm infants fed donor milk following the conversion back to their own mother’s milk, the leptin level increased almost 50% from the baseline to the highest value of 1774 pg/mL [120].

### 3.3. Resistin

So far, four reports have presented the data on the resistin level for two different biological samples [77,124,125,126], and one simultaneously characterized the resistin status in three different biological fluids: maternal and cord blood and placenta [113] (Table 3).

In 2006, Cho’s group [125] reported that the cord serum resistin level was 21.3 ± 1.1 ng/mL, and it was significantly higher than for maternal serum (10.1 ± 1.1 ng/mL). Moreover, their findings showed that the umbilical serum resistin level was positively correlated with the resistin level in maternal serum, and it was negatively correlated with the newborn’s birth weight. Similar results, albeit with slightly lower values, were presented by Floeck’s group [126], namely 17.7 ng/mL for umbilical cord and 8.0 ng/mL for maternal plasma. Moreover, the newborn and maternal resistin concentrations were positively associated with gestational age (r = 0.3 and r = 0.3) but not with the maternal BMI, the mode of delivery, pre-existing maternal diseases (diabetes mellitus, preeclampsia, and hypothyroidism), or the anthropometric parameters of infants.

Only two reports have presented the results for the maternal serum resistin level and placenta tissue [77,124]. Haugen and coworkers [77] analyzed the level of resistin in placental tissue and maternal plasma in relation to the occurrence of preeclampsia and reported that pre-eclamptic mothers plasma’s resistin concentration (5.7 ± 0.4 ng/mL) was significantly higher than for physiological pregnancy (4.7 ± 0.3 ng/mL). Nevertheless, for the placental tissue samples, no significant difference in the resistin mRNA levels between the analyzed cohorts was found [77].

Subsequent research [124] confirmed the previously observed relationships, although the noted resistin levels for maternal and cord plasma were slightly lower than those reported previously [77]. As demonstrated in the cross-sectional study by Erol’s group [124], the resistin level for pre-eclamptic mothers was significantly higher (mild preeclampsia group: 3.3 ± 0.6 ng/mL, severe preeclampsia group: 3.8 ± 0.4 ng/mL) than for healthy mothers (2.6 ± 0.4 ng/mL). Similarly to maternal serum, the placental tissue samples of severe pre-eclamptic mothers showed a significantly higher resistin expression compared to mild pre-eclamptic mothers and the normal cohorts. Moreover, the maternal serum resistin concentration and the placental resistin expression level were negatively correlated with birth weight (r = −0.5 and r = −0.3, respectively) (Table 3).

In 2018, Shang’s group [113] analyzed the resistin level in women with GDM and their newborns in relation to oxidative stress and noted that the level of resistin in maternal blood (GDM: ~18.0 ng/mL and non-GDM: ~12.0 ng/mL) was higher than in cord plasma (GDM: ~14.0 ng/mL and non-GDM: ~12.0 ng/mL) and placenta (GDM: ~12.0 ng/mg placenta and non-GDM: ~9.0 ng/mg placenta). The resistin levels in maternal and cord plasma and placenta were significantly higher for GDM mothers in comparison to the non-GDM cohort. In the same study, the impact of macrosomia on the resistin level in maternal and cord plasma and placenta was analyzed [113].

The level of resistin in cord plasma was significantly higher for mothers who delivered a macrosomic baby (35.8 ± 3.2 ng/mL) in comparison to normal birth weight newborns (28.2 ± 5.2 ng/mL). On the other hand, the placenta samples and maternal plasma did not show significant differences between the analyzed cohorts. The resistin level in maternal plasma was 31.1 ± 6.8 ng/mL for the macrosomic group and 30.0 ± 6.3 ng/mL for normal weight, and for the placental samples, 22.95 ± 5.25 ng/mg and 22.0 ± 6.3 ng/mg, respectively [113].

### 3.4. Irisin

The data on the irisin level include ten reports for two different biological samples [128,129,130,131,132,133,134,135,136,137] and one that simultaneously characterizes the resistin status in three different biological fluids: maternal and cord blood and placenta [127] (Table 3). Interestingly, so far only one study has recorded the concentrations of irisin in neonate blood and maternal milk [129] and three in maternal serum and milk [130,133,136].

The study by Aydin’s group [130] demonstrated that the milk irisin level is not related to the milk maturation stage, being about 550 ng/mL for colostrum, 520 ng/mL for transitional, and 460 ng/mL for mature milk; however, for mothers with GDM, the concentration was significantly lower in colostrum and transitional milk (350 ng/mL and 410 ng/mL, respectively) than in the respective samples of non-GDM mothers. Additionally, irisin plasma from lactating women with GDM in the colostral and transitional milk period (250 ng/mL and 360 ng/mL, respectively) was significantly lower than from normal lactating women’s plasma (approximately 520 ng/mL and 500 ng/mL, respectively) [130]. Later studies showed higher irisin values for maternal serum (662.7 ± 169.5 ng/mL) but lower for colostrum [133]; namely, the colostral irisin level (191.0 (641.9–30.7) ng/mL) was approximately three times lower than in the respective maternal serum, and more than two times lower than was reported previously by Aydin and coworkers [130]. The authors [133] did not find a relationship between the colostral and maternal irisin levels. In 2019, Fatima’s group [136] evaluated the irisin levels in maternal serum (at the 28th week of gestation and at the 6th week postpartum) and in colostrum and the mature milk of GDM and non-GDM mothers. The recorded values were lower than the previously published data [130,133]. Fatima and coworkers [136] reported that for both maternal serum at the 28th week of gestation and at 6 weeks postpartum, the irisin levels of GDM mothers were significantly lower than for the non-GDM cohort (GDM: 42.1 ± 3.2 pg/mL and non-GDM: 72.9 ± 9.1 pg/mL, respectively, and GDM: 138.3 ± 6.8 pg/mL and non-GDM: 265.0 ± 40.9 pg/mL, respectively). The same pattern was observed for the milk samples from the successive stages of lactation; namely, the colostrum and mature milk irisin levels of GDM mothers were significantly lower than for the non-GDM cohort (GDM: 10.4 ± 4.7 pg/mL: and non-GDM: 57.1 ± 8.3 pg/mL, respectively, and GDM: 15.4 ± 0.4 pg/mL and non-GDM: 56.4 ± 9.6 pg/mL, respectively). However, these levels were much lower than those reported previously. Additionally, the maternal serum irisin level was positively associated with the maternal BMI, and also the colostral and mature milk irisin concentrations were positively correlated with the offspring’s weight at the sixth week postpartum [136] (Table 3).

Ebert and coworkers [127] noted that the levels of irisin in maternal serum prepartum (before elective cesarean section) and postpartum as well as in cord serum were at almost the same level (268.8 ng/mL, 260.9 ng/mL, and 246.5 ng/mL, respectively), and for the first time, reported the placental irisin content per total placental protein (53.3 μg/g total protein) [127]. Additionally, the same group analyzed the irisin levels in diabetic mothers at standard pregnancy glucose testing and reported the same level for GDM and gestational age-matched normal mothers (482.1 ng/mL and 466.6 ng/mL, respectively). However, in the postpartum period irisin level, it was significantly higher in mothers with prior GDM (446.3 ng/mL) than for the control group (378.0 ng/mL) [127]. At the same time, Yuksel and coworkers [131] investigated the relationship between maternal and cord blood irisin in relation to maternal glycemic status and found that the irisin level in the serum of GDM mothers was significantly lower than in the control cohort (258.3 ± 127.9 vs. 393 ± 178.9 ng/mL, respectively). In contrast, the irisin level in cord serum was not associated with maternal glycemic status (GDM: 357.2 ± 248.0 and non-GDM: 333.2 ± 173.4 ng/mL, respectively) [131]. Wawrusiewicz-Kurylonek’s group [128] analyzed the maternal serum irisin levels between 24 and 28 weeks of pregnancy and reported that pregnant women with GDM had a markedly lower irisin concentration than the normoglycemic (non-GDM) cohort (1679 (1308–2171) ng/mL vs. 1880 (1519–2312) ng/mL, respectively) [128]. The continuation of studies in the perinatal period showed that both the maternal and cord serum irisin levels at term delivery were at the same level regardless of glycemic status: for mothers with GDM values, they were 1524 (1261–1783) ng/mL vs. 1723 (1460–1988) ng/mL, respectively; and for mothers with non-GDM, the values were 1375 (1084–1652) ng/mL vs. 1257 (1153–1415) ng/mL, respectively. However, the cord serum irisin level was significantly lower than maternal serum, but for non-GDM groups only. Moreover, three months after delivery, the maternal irisin levels were lower than at birth for both the GDM and non-GDM cohorts (1109 (841–1495) ng/mL vs. 1137 (822–1372) ng/mL, respectively) [128].

Hernandez-Trejo’s group [132], apart from maternal and cord serum cytokines, analyzed the irisin status in the mother–child dyad and reported that the concentration of maternal irisin (151.4 ± 127.0 ng/mL) was significantly higher than in the cord sample (94.8 ± 77.1 ng/mL), but for both, no correlations with the offspring anthropometric data were found [132].

Foda and Foda [134] analyzed the irisin levels in maternal and cord serum in relation to the mode of delivery and the occurrence of preeclampsia [134]. Their findings revealed that the irisin level in mildly pre-eclamptic mothers regardless of the delivery mode, namely for vaginal delivery (at birth: 726.3 ± 102.6 ng/mL and after delivery: 835.0 ± 98.0 ng/mL, respectively) and elective c-section (at birth: 629.9 ± 107.1 ng/mL and after delivery: 676.6 ± 99.5 ng/mL, respectively) were significantly lower than that of normal pregnancies vaginally delivered (at birth: 914.0 ± 90.3 ng/mL and after delivery: 975.9 ± 63.7 ng/mL) [134]. For elective cesarean sections, the irisin levels in cord serum for normal pregnancies (97.1 ± 15.7 ng/mL) and with mild preeclampsia (97.3 ± 1 6.6 ng/mL) were significantly lower than for the cohort of mild preeclampsia for vaginal delivery (120.9 ± 14.81 ng/mL), and additionally, the irisin level in cord serum correlated with the birth weight of newborns [134] (Table 3). In 2018, Pavlova’s group analyzed the irisin plasma levels in maternal and cord plasma samples in relation to gestational age (preterm and term birth) [135]. The level of irisin in maternal plasma delivered early preterm was 12.0 ± 2.4 ng/mL and did not differ significantly in comparison to mothers who delivered at term (11.5 ± 1.5 ng/mL). Similarly, no significant differences were observed for cord plasma samples in relation to the analyzed cohorts (early preterm vs. term birth: 7.7 ± 2.2 ng/mL vs. 6.8 ± 1.5 ng/mL, respectively). In turn, Ersahin and Yurci [137] analyzed irisin levels in relation to maternal glycemic status and found that in the serum of GDM mothers, the irisin level was lower (5.3 ± 0.4 μg/mL) than in healthy women: 7.7 ± 4.5 μg/mL. On the other hand, no significant differences were noted for the cord blood irisin level between the analyzed cohorts (GDM: 4.9 ± 3.1 μg/mL and non-GDM: 5.0 ± 2.1 μg/mL, respectively) [137]. Their findings revealed a positive relationship between the irisin level and BMI of GDM mothers, but no correlation was observed for the maternal irisin concentration and cord blood or between the neonate birth weight and maternal/cord serum in the GDM cohort [137].

Mól and coworkers [129] compared the irisin levels in newborns’ serum and maternal milk at the first and fifth week postpartum of very low birth weight (VLBW) preterm and full-term newborns. The authors stated that the prematurity of neonates translated to a significant decrease in serum irisin, namely (approximately 1.5 μg/mL at the first week and 2.3 μg/mL at the fifth week, respectively) in comparison to the term offspring (approximately 2.2 μg/mL at the first week and 3.5 μg/mL at the fifth week, respectively). Their findings demonstrated that at the first week postpartum, neonatal irisin positively correlated with both birth weight and length as well as head circumference [129]. In contrast, prematurity did not significantly influence the milk irisin level (approximately 2.9–3.6 μg/mL) at the first week postpartum, although the concentration was higher than in neonatal serum regardless of gestational age [129].

### 3.5. Ghrelin

So far, only three reports have presented the data on the ghrelin level for two different biological samples [121,138,139], and none has simultaneously characterized ghrelin status in three different biological fluids (Table 3).

Gómez-Díaz’s group [139] analyzed the ghrelin levels in maternal and cord blood samples and detected significantly lower maternal plasma ghrelin levels for term-delivered mothers regardless of diabetes type (GDM: 273 pg/mL and T2DM: 239 pg/mL, respectively) in comparison to non-diabetic mothers (439 pg/mL) [139]. They pointed out that the ghrelin concentration in maternal plasma differs significantly in relation to the gestation week for GDM but not for T2DM mothers; i.e., the ghrelin level was higher for preterm-delivered GDM mothers in comparison to term-delivered GDM mothers (451 pg/mL and 128 pg/mL, respectively) [139]. On the other hand, the ghrelin concentration in cord plasma was twice as high as in term-delivered maternal plasma, but it remained constant irrespective of the glycemic status of the mother (GDM: 872 pg/mL, T2DM: 832 pg/mL, and non-diabetic mothers: 889 pg/mL, respectively), and additionally, in contrast to diabetic maternal plasma, irrespective of the delivery week [139].

The subsequent studies by Kimber-Trojan’s group showed completely different values of the maternal and cord serum ghrelin concentrations: 933 pg/mL for maternal and 19.5 pg/mL for cord serum [121]. Additionally, they investigated the maternal and cord serum ghrelin levels of term-delivered women who gained excessive gestational weight (EGWG) and observed a significant increase in the cord ghrelin level for EGWG mothers (525 pg/mL) in comparison to the control cohort (19.5 pg/mL). For EGWG maternal serum, the ghrelin level was also higher, but not significantly (1187 pg/mL and 933 pg/mL, respectively).

So far, only Shimizu’s group [138] has analyzed the ghrelin level for the same patient but at the successive stages of life, namely in the cord and neonate plasma samples of very preterm and VLBW infants. In comparison to cord serum and the newborn’s serum immediately after birth (approximately 4–5 fmol/mL), the ghrelin level of neonates from 2 to 8 weeks old was significantly higher (approximately 15 fmol/mL), though it remained at an almost constant level [138].

### 3.6. Nesfatin-1

Only two reports have presented the data on the nesfatin-1 level for two different biological samples [140,141], and none has simultaneously characterized the nesfatin status in three different biological fluids (Table 2).

In 2010, Aydin and coworkers [140] documented the nesfatin-1 levels in the maternal serum, colostrum, and mature milk of GDM mothers, although for very small cohorts (GDM: 10 and non-GDM: 10 samples, respectively). The authors found that in the serum of lactating women, for both analyzed groups, namely non-GDM and GDM, the nesfatin-1 levels during the colostral period were 0.9 ± 0.4 ng/mL and 0.7 ± 0.2 ng/mL and were lower than at the second week of lactation: 1.1 ± 0.3 ng/mL and 0.9 ± 0.2 ng/mL, respectively [140]. Similar results were obtained for the colostrum 0.8 ± 0.3 ng/mL and mature milk 1.0 ± 0.3 ng/mL of GDM mothers. In contrast to the GDM cohort, the milk nesfatin-1 level of non-GDM mothers was higher at the beginning of lactation (1.6 ± 0.2 ng/mL) than for mature milk (1.2 ± 0.4 ng/mL). Additionally, the maternal serum nesfatin-1 levels of hyperglycemic mothers were lower than for normoglycemic women, regardless of the lactation stage [140].

A later study by Aslan and coworkers [141] analyzed the nesfatin-1 levels in maternal and cord serum in relation to maternal glycemic status, but a bigger cohort of samples (30 samples from GDM and 30 from non-GDM) was enrolled. The authors found that the maternal nesfatin-1 levels were significantly lower in women with GDM (5.5 ± 8.1 ng/mL) compared with non-GDM pregnant women (8.1 ± 23.9 ng/mL). In contrast to maternal serum, the cord blood nesfatin-1 level was not affected by glycemic status (5.4 ± 4.0 ng/mL for GDM and 6.2 ± 10.3 ng/mL for non-GDM, respectively). Moreover, neither the maternal serum nor cord blood nesfatin-1 levels correlated with the neonatal birth weight (Table 3) [141].

### 3.7. Vaspin

The evidence concerning the vaspin level in maternal–infant dyads is fragmentary and includes one report only, which presents the data for two different biological samples [142] (Table 3).

Huo and coworkers [142] reported vaspin concentration in the serum of GDM mothers and its expression in the placenta. Before cesarean section, the level of vaspin in hyperglycemic maternal serum was significantly lower (0.5 ± 0.2 ng/mL) than in the normoglycemic group (0.8 ± 0.3 ng/mL). However, at the third day after childbirth, a significant decrease to 0.4 ± 0.1 ng/mL for the GDM cohort but not for the normo-glycemic group (0.7 ± 0.3 ng/mL) was observed [142]. On the other hand, the placental mRNA vaspin levels were similar for the GDM and non-GDM cohorts (0.6 ± 0.3 and 0.7 ± 0.3, respectively), whereas for GDM mothers, mRNA vaspin expression negatively correlated with the newborns’ birth weight (r = –0.5).

### 3.8. Visfatin

The literature concerning the visfatin level in maternal–infant dyads include five reports, which present the data for two different biological samples [122,123,129,144,145] and one for three different biological fluids, namely maternal serum, umbilical cord blood, and placental tissue [143] (Table 3).

Malamitis-Puchner and coworkers [144] reported that the visfatin levels in maternal and cord serum were at similar levels, namely 18.8 ± 34.3 ng/mL and 19.4 ± 4.9 ng/mL, respectively, and a strong correlation between these values was found (r = 0.7). On the other hand, no correlations of serum visfatin levels with the gender, mode of delivery, and parity were observed [144]. In 2019, Lu and coworkers [143] analyzed the relationships between visfatin and GDM and observed significantly higher maternal serum visfatin concentrations for both analyzed sub-cohorts, namely diet (G1)- and insulin (G2)-treated GDM (50.7 ± 14.2 ng/mL and 48.5 ± 14.5 ng/mL, respectively) in comparison to the non-GDM cohort (31.1 ± 7.5 ng/mL). Similar relationships, though with lower values, were observed for the cord visfatin level, namely 35.4 ± 10.4 ng/mL for GDM G1, 36.7 ± 12.0 ng/mL for GDM G2, and 21.0 ± 5.8 ng/mL for the non-GDM cohort, respectively. For placenta tissue, visfatin’s expression (IOD/area) in the GDM cohort was significantly higher than for the non-GDM group, but no significant difference between the G1 and G2 sub-cohorts (52.7 ± 17.3 for GDM-G1, 52.5 ± 9.5 for GDM-G2, and 31.8 ± 8.3 for non-GDM, respectively) was found [143]. A later study by Pekal’s group [122] presented the visfatin level in the maternal and cord serum in relation to the newborn’s weight at birth and showed significantly higher cord visfatin levels for both groups, namely SGA and LGA (5.84 ng/mL and 5.69 ng/mL, respectively) than in AGA (3.20 ng/mL). In contrast to the cord, the maternal serum visfatin level was significantly lower for SGA (1.6 ng/mL) and LGA (1.7 ng/mL) than for the AGA (4.4 ng/mL) cohort. However, no associations between maternal and cord serum and the newborn’s anthropometric measurements such as birth weight (Table 3) were detected [122]. The latest report by Gök and coworkers [123] noted that the maternal and cord visfatin levels were significantly higher for pre-eclamptic mothers (3.0 ng/mL and 3.6 ng/mL, respectively) in comparison to the normal pregnancy group (0.6 ng/mL and 1.0 ng/mL, respectively), and additionally, positive correlations between the maternal and cord serum visfatin levels and maternal BMI for both cohorts were found [123].

Bienertová-Vašků and coworkers [145] analyzed the visfatin levels in maternal serum and milk up to the sixth month of lactation and observed that the milk visfatin level (854–1851 ng/mL) was approximately 100 times higher than in maternal serum (2.5–10.7 ng/mL). Additionally, for maternal serum’s visfatin level, a strong decrease as lactation progressed was observed from 10.7 ± 9.1 ng/mL at birth to 2.5 ± 1.1 ng/mL at the sixth month postpartum, whereas such a trend was not observed for milk [145].

On the other hand, Mól’s group [129] described the visfatin levels in newborn serum and maternal milk at the first and fifth week postpartum in relation to the birth week. In the first week of lactation, the milk visfatin level reached approximately 8–10 ng/mL regardless of the week of delivery and did not change significantly in the fifth week postpartum. The newborn serum visfatin level for the VLBW preterm cohort at the first week postpartum was approximately 6.5 ng/mL and was significantly lower in comparison to the full-term cohort (~8.5 ng/mL). At the fifth week postpartum, an opposite trend for both analyzed groups was observed: for the preterm group, an increase to the value ~7.5 ng/mL, and for the term group, a decrease in visfatin concentration (~6.5 ng/mL) were noted [129]. Moreover, the newborn serum visfatin levels for both groups at the corresponding lactation stages were lower than in maternal milk. However, at the fifth week postpartum, newborn visfatin was negatively associated with the anthropometric parameters, namely birth length and weight as well as head circumference [129].

### 3.9. Chemerin

The analysis concerning the chemerin level in maternal–infant dyads is fragmentary. So far, only two reports present the data on the chemerin level for two different biological samples (maternal and cord plasma) [146,147].

Barker and coworkers [146] assessed the chemerin levels in maternal and cord samples in relation to the presence of GDM and maternal obesity and determined that neither maternal obesity nor gestational diabetes influenced the plasma chemerin level (125.7 ± 7.4 ng/mL for lean; 115.1 ± 5.9 ng/mL for overweight; and 128.3 ± 6.6 ng/mL for obese mothers, respectively, and 117.6 ± 3.5 ng/mL for GDM and 124.2 ± 4.0 ng/mL for non-GDM mothers, respectively). However, for cord plasma, a significantly higher level of chemerin in the obese cohort (141.6 ± 7.7 ng/mL) in comparison to the lean group (115.0 ± 8.3 ng/mL) but not in relation to the glycemic status of the mother was reported. Additionally, for the first time, the presence of chemerin mRNA in both placental and adipose tissues was demonstrated; nevertheless, its level was similar regardless of maternal obesity and glycemic status (approximately 9 ng/mg of protein for non-obese mothers and 10 ng/mg of protein for obese mothers) [146].

In 2019, Ustebay and coworkers [147], for the first time, demonstrated the presence of chemerin in maternal milk (colostrum: range 25–35 ng/mL, transitional: approximate range 20–23 ng/mL, and mature milk: approximate range 10–20 ng/mL); its level was significantly higher than for maternal plasma (approximately range 8–25 ng/mL). Moreover, for GDM and DM mothers, the chemerin concentrations in the milk from the successive stages of lactation, namely colostrum, transitional, and mature milks, and its respective maternal plasmas were significantly higher than for the control cohorts. Additionally, with lactation progression, regardless of the maternal glycemic status for both the milk and respective maternal plasma samples, the chemerin level gradually decreased [147].

### 3.10. Apelin

Only three reports present the data on the apelin level for two different biological samples [141,149,150], and one simultaneously characterizes the apelin status in three different biological fluids [148] (Table 3).

Aslan and coworkers [141] analyzed the apelin levels in maternal and cord serum in relation to the maternal glycemic status. The apelin level in GDM maternal serum was significantly higher (13.5 ± 8.3 ng/mL) than in non-GDM mothers: 9.6 ± 5.9 ng/mL. In contrast to maternal serum, the cord apelin concentrations were at similar levels regardless of the maternal glycemic status (GDM: 8.8 ± 4.3 ng/mL and non-GDM: 8.2 ± 1.9 ng/mL, respectively). Additionally, the negative correlations between both the cord and maternal serum apelin levels and birth weight and gestational age were found [141] (Table 3).

In contrast, Oncul and coworkers [149] reported a significantly lower apelin level for GDM cord plasma (0.1 ± 0.03 ng/mL) than for the non-GDM cohort (0.3 ± 0.1 ng/mL). For GDM, the maternal plasma apelin level was also lower (0.1 ± 0.05 ng/mL) but not significantly in comparison to non-GDM mothers (0.2 ± 0.1 ng/mL).

A later study by Marousez and coworkers [150] analyzed the maternal plasma and breast milk apelin levels in relation to maternal GDM and obesity and demonstrated that the plasma apelin levels were significantly affected; namely lower levels were observed for obese mothers with GDM (~0.15 ng/mL) and without GDM (~0.19 ng/mL) than for non-obese mothers (~0.24 ng/mL). In contrast to maternal plasma, the breast milk apelin levels increased with maternal GDM and BMI (non-obese: ~12 ng/mL, obese: ~25 ng/mL and obese with GDM: ~30 ng/mL, respectively). Moreover, the milk and plasma apelin levels significantly correlated with preconceptional BMI, but not with the infant’s birth weight [150] (Table 3).

In 2022, Hanssens and coworkers [148] focused on the maternal, cord, and placental apelin levels in relation to maternal obesity. For the cord plasma samples, significantly lower levels were observed for obese (0.2 ng/mL) than for non-obese (0.4 ng/mL) mothers; however, such a relationship was not observed for the maternal apelin levels in non-obese (approximately 0.3 ng/mL) and obese (approximately 0.3 ng/mL) mothers. In contrast, the placental tissue apelin expression was higher for obese than for the non-obese cohorts (obese: 1.3 AU and non-obese: 1.0 AU, respectively) [148].

### 3.11. Adropin

The analysis concerning the adropin level in maternal–infant dyads is not comprehensive and includes five reports which present the data for two different biological fluids only [115,130,133,151,152] (Table 3).

Celik and coworkers [151] analyzed the adropin level in maternal and cord serum in relation to the glycemic status of the mother. For both maternal (2.4 ± 2.0 ng/mL) and cord blood (1.5 ± 0.9 ng/mL), significantly lower adropin levels were observed in the GDM cohort in comparison to the non-diabetic control group (3.3 ± 1.3 ng/mL and 3.3 ± 1.3 ng/mL, respectively). Additionally, in the GDM cohort, the cord plasma adropin level negatively correlated with the pregnancy duration and was related to the delivery mode [151].

In contrast, Aydin’s group [115] analyzed the adropin level in IUGR neonates and their mothers, where the maternal and cord serum adropin levels were significantly lower for IUGR (2.6 ± 1.8 ng/mL and 1.8 ± 0.8 ng/mL, respectively) in comparison to the normal group (5.9 ± 5.9 ng/mL and 2.8 ± 1.6 ng/mL, respectively). Additionally, a negative correlation of the maternal adropin level with BMI but not with the neonatal birth weight was found [115].

On the other hand, Cakmak and coworkers [152] analyzed the maternal and cord adropin levels in relation to preeclampsia for both the maternal and cord samples, and significantly lower adropin levels (71.2 ± 22.2 ng/L and 92.4 ng/L, respectively) for the preeclampsia group were noted in comparison to the normal group (100.8 ± 27.0 ng/L for maternal and 106.2 ng/L for cord, respectively). Moreover, the cord adropin levels were weakly correlated with the gestational age at delivery (r = 0.3) and the neonatal birth weight (r = 0.3) [152].

Aydin and coworkers [130], for the first time, analyzed the adropin level in the milk samples during the lactation progression and parallelly in the corresponding maternal plasma of lactating women [130]. For the normal cohort adropin levels in colostrum, transitional and mature milk remained at the same level similarly to the respective maternal plasmas (~15–18 ng/mL for milk and ~16 ng/mL for maternal plasma, respectively). However, for the plasma of GDM lactating mothers, the level of adropin was significantly lower during the colostral and transitional periods than for the non-GDM group (~8 ng/mL GDM for colostral period, ~6 ng/mL GDM for transitional milk period, and ~13 ng/mL non-GDM for colostral period, ~10 ng/mL non-GDM for transitional milk period, respectively). Moreover, the colostrum adropin level for GDM mothers (~7.5 ng/mL) was significantly lower than for non-GDM mothers (~17 ng/mL) [130].

The concentration of colostral adropin reported by Briana’s group [133] was at a similar level as was previously demonstrated (13.7 (30.3–5.2) ng/mL), whereas in maternal serum (2.1 ± 0.7 ng/mL), in contrast to the earlier report, it was significantly lower. Additionally, a positive correlation between the colostrum and maternal serum adropin concentrations was found (r = 0.4) [133]

### 3.12. Copeptin

The analysis concerning the copeptin level in maternal–infant dyads is incomplete and includes five reports which present the data for two different biological fluids only [130,133,153,154,155] (Table 3).

Foda and Aal [153] evaluated the copeptin level in relation to the mode of delivery. During delivery, the copeptin level was significantly higher (291.7 ± 90.5 pg/mL) than during pregnancy (115.1 ± 41.6 pg/mL). Moreover, that relationship was observed for all types of delivery, although for vaginal delivery, the copeptin level was higher (443.9 ± 116.2 pg/mL) than for elective and intra-partum CS (215.1 ± 41.1 pg/mL and 306.0 ± 73.0 pg/mL, respectively). In contrast, the cord copeptin levels were not related to the delivery mode and were at comparable levels for vaginal delivery (96.4 ± 18.2 pg/mL) and for elective (80.8 ± 11.6 pg/mL) as well as intra-partum CS (87.4 ± 24.4 pg/mL) [153].

On the other hand, Blohm and coworkers [154] found that the cord copeptin level was 10 times higher than in maternal blood: 103.4 ± 22.9 pmol/L and 10.4 ± 1.7 pmol/L, respectively.

Ulu and coworkers [155] investigated the copeptin level in maternal and cord serum in relation to fetal distress. The copeptin levels for both were at a similar level regardless of fetal status: for the intrapartum fetal distress group, maternal was 9.5 ± 2.0 ng/mL and cord was 8.8 ± 1.4 ng/mL; and for the control cohort, maternal was 9.3 ± 0.8 ng/mL and cord was 8.0 ± 4.4 ng/mL. Moreover, a positive correlation between the maternal and cord copeptin levels, but not with birth weight, was detected [155].

The presence of copeptin in maternal milk at successive maturation stages and in maternal plasma was demonstrated for the first time by Aydin and coworkers [130]. The maternal plasma (non-GDM: 7–8 ng/mL and for GDM: 8–12 ng/mL) and milk copeptin concentrations did not vary significantly with the lactation progression, namely in colostral (non-GDM: ~9 ng/mL; GDM: ~14 ng/mL), transitional (non-GDM: ~8 ng/mL; GDM: ~11 ng/mL), and mature milk periods (non-GDM: ~9 ng/mL; GDM: ~10 ng/mL). However, during the colostral period, the copeptin levels for both the maternal plasma and colostrum of lactating diabetic mothers were significantly higher compared with the normoglycemic cohort (for plasma ~12 ng/mL for GDM and ~8 ng/mL for non-GDM, and for colostrum, ~14 ng/mL for GDM and ~9 ng/mL for non-GDM) [130].

Further research by Briana’s group [133] showed much lower values for the copeptin levels: the colostral copeptin level was 0.6 [2.4–0.4] ng/mL and its concentration was approximately double that of maternal serum (0.3 ± 0.1 ng/mL). Moreover, a weak positive correlation between the colostrum and maternal serum copeptin levels was found (r = 0.3) [133].

### 3.13. Omentin

Only two reports present the data on the omentin levels for maternal and cord plasma [156,157] (Table 3).

Barker and coworkers [156] analyzed the level of omentin in the maternal and cord plasmas, placenta, and adipose tissue of women with obesity delivered at term via cesarean section [156] and ascertained that pre-existing obesity in both normoglycemic and GDM mothers had a significant impact on the maternal omentin level. Normoglycemic obese mothers (7.1 ± 0.9 ng/mL) had significantly lower plasma omentin levels in comparison to normoglycemic mothers with normal weight (19.5 ± 2.3 ng/mL) and similarly obese women with GDM (8.2 ± 1.2 ng/mL) in comparison to the GDM-non-obese cohort (12.1 ± 1.4 ng/mL). Barker’s group found that the maternal omentin level at delivery negatively correlated with neonatal weight with no significant impact of obesity and GDM on the cord omentin level [156]. Later studies by Franz’s group [157] demonstrated no effect of diabetes on the maternal omentin level at 32 weeks of gestation, at the standard time for gestational diabetes testing (118 ± 77 ng/mL for GDM and 150 ± 89 ng/mL for non-GDM, respectively); however, the higher BMI of mothers was associated with a lower omentin concentration. The level of cord omentin was significantly lower for diabetic mothers than for normoglycemic mothers (106 ± 61 ng/mL and 134 ± 45 ng/mL, respectively), but no correlations with the delivery mode and birth weight were found [157].

### 3.14. Dermicidin

So far, only one report by Ustebay and coworkers has characterized the dermicidin level for two different biological fluids [147] (Table 3).

The presence of dermicidin in maternal milk was demonstrated for the first time by Ustebay’s group [147]. The dermicidin concentrations, regardless of the milk maturation stage (colostrum: ~70–75 ng/mL, transitional: ~45–55 ng/mL, and mature milk: ~30–40 ng/mL), were much higher than for the respective maternal plasma (~20–25 ng/mL). Moreover, the milk dermicidin levels of GDM and DM mothers from the successive stages of lactation, namely colostrum, transitional, and mature milks, and the respective maternal plasmas, were significantly higher than for the control cohorts. Additionally, for both the milk and respective maternal plasma samples, the dermicidin concentration decreased with the lactation progression regardless of the maternal glycemic status [147].

## 4. Discussion

Some of the main lifestyle diseases are metabolic disorders that directly affect the longevity and quality of adult life and also affect the group of women at the reproductive age [158,159,160,161]. In recent years, the hypothesis of metabolic programming during fetal development has become increasingly prominent [44,162]. In line with the hypothesis, many metabolic diseases in later life may be the result of the adaptation of developing fetuses to an altered intrauterine environment triggered by maternal energy imbalances and/or metabolic disorders during pregnancy. Among the most frequent are the obesity of women and a hyperglycemic state identified for the first time during pregnancy [162,163,164], namely GDM, which is one of the factors responsible for the modification of the intrauterine environment, and thus, exerts effects on the offspring and also maternal health. The increasing prevalence of metabolic disorders globally calls for greater scientific attention to their effects on the wellbeing of the mother and child and also on the complex relationship in the maternal–infant dyad [164]. The analyzed reports confirm the suggestions that various adipokines participate in metabolic events related with pregnancy pathologies and can translate to the lactation period, namely the alterations associated with maternal gestational diabetes, preeclampsia, and the disorders of intrauterine growth [112,165]. The adiponectin, leptin and resistin alternations translate to the worsening of maternal insulin resistance as well as metabolic stress and, in consequence, the promotion of an inflammatory environment, altered placenta functions, and finally, unfavorable conditions for the developing fetus [113,166]. However, the detailed mechanisms by which adipokines affect metabolic changes so far are not clearly identified and need to be explained.

This review summarizes the findings of 47 studies concerning the adipokine levels in at least two different biological fluids of maternal–infant dyads and includes all possible different sets of biological fluids: maternal and cord plasma, neonate/infant plasma, milk, and placental tissues, for both physiological and disturbed pregnancy and lactation. The current state of knowledge clearly indicates the possible impact of the maternal hyperglycemic status [113,118,131,139,143,146,148,151], preeclampsia [77,123,124,134,152], preterm delivery [119], and fetal growth restrictions [115,122] on the quantitative profile of adipokines important for energy homeostasis, the regulations of metabolism, and fat tissue.

On the basis of the obtained results, attempts were also made to determine the interdependence between the concentrations of adipokines. Among 14 analyzed adipokines for 13 relationships with different biological fluids were identified (the only exception was dermcidin, for which the selected study did not investigate such relationships), mostly in relation to the maternal health status and perinatal factors, such as the delivery week and neonatal birth weight. In almost all analyzed adipokine patterns in mother–infant dyads, the most relationships were investigated for maternal and cord plasma (29 out of 47) (Table 2). Only 4 out of the 47 selected studies investigated the relation of the adipokine patterns in maternal plasma and milk [28,133,136,150], with three in maternal plasma and placental tissue [77,124,142], two in cord plasma and neonatal/infant plasma [114,138], and two in neonatal/infant plasma and maternal milk [120,129]. For 7 out of the 47 studies, the relationships between the adipokine concentration and the perinatal risk factors were not investigated [116,130,140,145,147,148,153].

The vast majority of available studies identified significant differences in the level of adipokines in maternal and/or cord plasma in relation to perinatal risk factors such as GDM, preterm delivery, fetal growth abnormalities, preeclampsia, and maternal obesity. The most intensively studied disorder is GDM in relation to maternal obesity. However, significant differences in the adipokine levels in comparison to normoglycemic mothers in maternal–infant dyads were reported only for some of them; namely, increased concentrations in both biological materials (maternal and newborn origin) were reported for leptin [113], resistin [113], visfatin [143], and apelin [141]. In contrast, decreased concentrations, but for maternal plasma only, were reported for adiponectin [113], irisin [128,130,131], ghrelin [139], nesfatin-1 [140,141], vaspin [142], and adropin [130,151]. However, no significant differences were reported for copeptin [130] and omentin [157]. Nevertheless, the evidence for irisin in GDM maternal plasma is confusing; some investigators [128,130,131] reported a decrease, whereas Eberts and coworkers (2014) noted an increased level. For nesfatin-1 [140,141] and adropin [130,151] the observations for maternal plasma are consistent, as decreases were noted for both adipokines, whereas for apelin, the data are inconsistent [141,149].

The levels of hormones such as adiponectin [77], leptin [77,123], resistin [77,124], and visfatin [123] in pre-eclamptic maternal plasma were significantly higher, but the same relationship for cord plasma was only reported for leptin and visfatin [123]. In contrast, for irisin [134] and adropin [152], a decrease in their levels was reported for pre-eclamptic maternal plasma and for cord plasma adropin [152]. For obese mothers, the available data are limited, yet an increase in leptin [118], decrease in omentin [156] and comparable levels of adiponectin [118] and chemerin [146] were reported, but with no impact on the cord serum levels (the exception is cord chemerin). On the other hand, the data concerning the apelin levels in the serum of obese mothers are confusing [148,150]. Additionally, one report documented change related to preterm delivery, namely the increased maternal and decreased cord serum leptin levels [119]. Meanwhile, for the maternal and cord irisin [136] levels, no changes related to premature birth were detected.

Improper weight gain of the fetus (IUGR, EGWG, SGA and LGA) also translates to the alterations in the adipokine profile. In contrast to GDM, for which the most significant changes in maternal plasma were recorded, for fetal growth abnormalities, the alterations in concentration are manifested mainly in cord plasma. For cord adiponectin (IUGR) [115], leptin (EGWG) [121], ghrelin (EGWG) [121], and visfatin (SGA and LGA) [122], an increase was recorded, but with no impact on the maternal serum levels except for visfatin [122]. In contrast, for cord plasma adropin (IUGR) [115] as well as for maternal serum, a decrease was noted. Additionally, for cord but not for maternal leptin [122], an increase for LGA and a decrease for SGA newborns were indicated.

Despite the fact that the vast majority of the included studies used the same methodology, namely a commercially available ELISA kit dedicated to the determination of adipokine concentration in different biological fluids (Table 3), the reported values differ significantly. This phenomenon may be affected, at least to some extent, by the quality of the test used and the number of biological samples analyzed in the individual studies. The size of the analyzed cohorts differs significantly among the included studies. The largest percentage of studies (21 of 47) included the cohorts of 20–50, followed by studies that enrolled 51–100 (13 out of 47), then cohorts of less than 20 (7 out of 47), and finally, the largest cohorts of more than 100 (6 out of 47).

The magnitude of differences in the measured concentrations for particular adipokines was variable. Namely, the values reported by various research teams for resistin [113,124], ghrelin [121,139], and nesfatin-1 [140,141] show a 2- to 9-fold variation in magnitude, while for adiponectin [113,116], leptin [119,121], visfatin [113,122], chemerin [146,147], apelin [141,148], adropin [151,152], copeptin [130,155], and omentin [156,157], they is a 20- to 50-fold variation in magnitude, and for leptin, an 80-fold variation in magnitude. The values presented for irisin are extremely variable, namely from 12 to 7740 ng/mL, with a 645-fold variation in magnitude [127,128,131,136,137]. The observed variations in the adipokine concentrations between some of the studies are likely the net result of the validation of the assay kits and/or different time points of the collection of the biological samples. In light of the above, the reference values of adipokines related with the stages of normal pregnancy should be established.

However, the main limitations of the current state of knowledge in the area of mother–child dyads are the impossibility of detailed comparisons related to the different types of biological material pairs in the mother–child system and the predominance of reports focusing on the determination of a single adipokine only. Additionally, different perinatal risk factors (gestational diabetes mellitus, preeclampsia, obesity, and hyperglycemic state, abnormal fetal development, and preterm delivery) and the different time points of sample collection are among the significant limiting factors. The main criterion for including articles in this study was the evaluation of the adipokine pattern in at least two different biological fluids of maternal and/or neonatal origin, namely maternal plasma and milk as well as cord and neonatal plasma, and additionally, placental tissue. Unfortunately, most of the available studies are restricted to the determination of adipokine patterns, usually only in a single type of biological material from the mother– child system. However, regardless of the number of biological fluids included in the analysis, the presented data are inconsistent, which was highlighted in the reviews published recently [92,112] as well as in this work. Due to the lack of detailed knowledge or huge gaps in some aspects, where the identification of the real relationships of the analyzed profile of adipokines is hindered, or even in some cases, simply impossible. In our opinion, only the dedicated collection of biological samples and the investigation of a panel of adipokines will make it possible to gather reliable data allowing for specific constructive conclusions to be made. Moreover, such parameters might be helpful for a comprehensive understanding of the shaping nutritional effect of the maternal–infant dyad and its disturbance by perinatal risk factors.

## 5. Conclusions

The conducted review of the literature allowed us to identify areas that still require intensive, well-planned research, taking into account the optimal number of samples in the study groups, with particular emphasis on the week of pregnancy ending and the lactation period. The analysis of adipokine patterns in the maternal–infant dyad is urgently necessary to understand the potential modulatory functions of fetal, neonatal, and infant metabolism by maternally-derived hormones transferred prenatally, perinatally, and postnatally. The identified knowledge gap in the area of the adipokine profile in the maternal–infant dyad requires a detailed complementation to allow a multidisciplinary panels of experts to make optimal decisions based on the proven scientific evidence in the field of dedicated perinatal care, especially in the case of high-risk newborns.

Additionally, as the current state of knowledge presented above indicates, detailed knowledge of milk hormones related to the regulation of the energy balance of the mother–child dyad is needed to support the activities aimed at promoting breastfeeding. The vast majority of children worldwide are not breastfed as recommended, despite strong evidence demonstrating the direct benefits of breastfeeding for the health and cognitive function of newborns and infants. Another important issue is the cost of not breastfeeding, which translates to the rapid increase in childhood obesity and diabetes observed in recent years. Breastfeeding and its promotion are therefore one of the simplest and, from the economic point of view, cheapest strategies in preventing the development of metabolic disorders.

## Figures and Tables

**Figure 1 nutrients-15-04059-f001:**
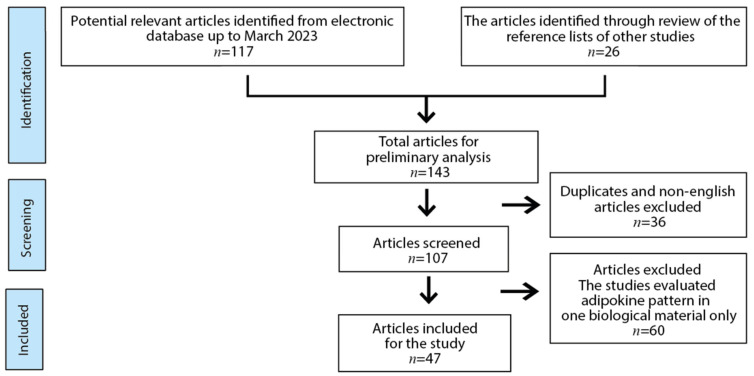
Flowchart for search strategy.

**Table 1 nutrients-15-04059-t001:** Scientific articles included in study.

Adipokine	The Number of Studies *n*/*N* (%)	References
Adiponectin	8/47 (17.0)	[28,77,113,114,115,116,117,118]
Leptin	11/47 (23.4)	[77,113,115,116,117,118,119,120,121,122,123]
Resistin	5/47 (10.6)	[77,113,124,125,126]
Irisin	11//47 (23.4)	[127,128,129,130,131,132,133,134,135,136,137]
Ghrelin	3/47 (6.4)	[121,138,139]
Nesfatin-1	2/47 (4.3)	[140,141]
Vaspin	1/47 (2.1)	[142]
Visfatin	6/47 (12.8)	[122,123,129,143,144,145]
Chemerin	2/47 (4.3)	[146,147]
Apelin	4/47 (8.5)	[141,148,149,150]
Adropin	5/47 (10.6)	[115,130,133,151,152]
Copeptin	5/47 (10.6)	[130,133,153,154,155]
Omentin	2/47 (4.3)	[156,157]
Dermcidin	1/47 (2.1)	[147]

*n*—number of studies, *N*—articles included in the study.

**Table 2 nutrients-15-04059-t002:** Adipokines examined in maternal–infant dyad.

Adipokines (*n*/*N*, %)	Maternal Plasma/Serum	Cord Blood Plasma	Neonatal/Infant Plasma/Serum	Milk	Placental Samples	References
Adiponectin (8/47, 17.0)	A	A				[117]
	A			A		[28]
						[116]
		A	A			[114]
	B	B				[118]
	A	A			A	[113]
	A				A	[77]
	D1	D1				[115]
Leptin (11/47, 23.4)	A	A			A	[113]
	C1				C1	[77]
			E	E		[120]
	B	B				[118]
	D	D				[115]
	A	A				[117]
	A	A		A		[119]
	D1	D1				[121]
	C2	C2				[123]
	E	A, C2, E				[122]
						[116]
Resistin (5/47, 10.6)	C3				C3	[77]
	A	A			A	[113]
	A	A				[125]
	E	E				[126]
	A				A	[124]
Irisin (11/47, 23.4)	A, E	A, E				[137]
	E	E				[135]
	A	A				[134]
	C2, C4	C2, C4				[131]
	A	A				[132]
	E	E				[128]
	C4	C4			E	[127]
			A	F1		[129]
	A			A		[136]
	E			E		[133]
						[130]
Ghrelin (3/47, 6.4)	D1	D1				[121]
		A	A			[138]
	E	A, E				[139]
Nesfatin-1 (2/47, 4.3)	E	E				[141]
						[140]
Vaspin (1/47, 2.1)	C1				A	[142]
Visfatin (6/47, 12.8)						[145]
	E	E	F2			[144]
			A	F1		[129]
						[143]
	E	A, C2, E				[122]
	C2	C2				[123]
Chemerin (2/47, 4.3)	C2, C4	C2, C4				[146]
						[147]
Apelin (4/47, 8.5)						[148]
	A, C2, C4	A, C2, C4				[141]
	E	E				[149]
	A, C2, C4			A, C2, C4		[150]
Adropin (5/47, 10.6)						[130]
	D2	D				[115]
	E			E		[133]
	C5	A				[152]
	E	E				[151]
Copeptin (5/47, 10.6)						[130]
	E			E		[133]
	E	E				[154]
	E	E				[155]
						[153]
Omentin (2/47, 4.3)	A	C2			C2	[156]
	C2	A				[157]
Dermcidin (1/47, 2.1)						[147]
negative relationship		positive relationship	no relationship		no information	

The relationship of adipokin concentration with: (A) weight of neonates/infants, (B) visceral/intra-abdominal fat (measure of adiposity), (C1) maternal triglycerides in the pathological condition, C2) maternal BMI, C3) maternal HDL cholesterol in the PE group, (C4) HOMA-IR, (C5) maternal blood pressure, (D) another adipokine in the IUGR group, (D1) umbilical cord SFRP5 in the EGWG group, (D2) endothelin in the IUGR group, (E) between two analyzed fluids, (F1) milk adipokine in the 1st and the 5th week of lactation, (F2) adipokine on the 1st and 4th day. Abbreviations: BMI—Body Mass Index, EGWG—Excessive Gestational Weight Gain, HOMA-IR—Homeostatic Model Assessment for Insulin Resistance, IUGR—Intrauterine Growth Restriction, PE—Preeclampsia, SFRP5—Secreted Frizzled-Related Protein 5 (adipokine).

**Table 3 nutrients-15-04059-t003:** Summary of studies evaluating adipokine pattern in maternal–infant dyad.

Adipokines	Maternal Plasma/Serum	Cord Blood Plasma	Neonatal/Infant Plasma/Serum	Milk	Placenta	Sample Size	Method	References
Adiponectin	13.5 ± 0.8 µg/mL	29.9 ± 3.1 µg/mL				*n* = 27 mothers; *n* = 23 infants	Immunoassays	[117]
	2nd trimester: 8.59 ± 6.54; µg/mL; 3rd trimester 7.64 ± 3.89 µg/mL; NS			0 month: 17.1 ± 8.8 ng/mL; 2nd month: 11.5 ± 8.5 ng/mL; *p* < 0.001		*n* = 155	ELISA	[28]
	Lifestyle interv. 7.8 µg/mL; Standard 8.0 µg/mL; NS	Lifestyle interv.22.4 µg/mL; Standard 22.3 µg/mL; NS				*n* = 1951 women; *n* = 1174 infants	RIA	[116]
		Total 14.9 µg/mL; Males: 15.1 µg/mL; Females: 14.6 µg/mL; NS	Total 9.2 µg/mL; Males 8.8 µg/mL; Females 10.4 µg/mL; NS			Cord *n* = 141; Children *n* = 40	ELISA	[114]
	GDM: ~0.5 µg/mL; non-GDM: ~0.8 µg/mL; *p* < 0.05	GDM: ~2.7 µg/mL; non-GDM: ~2.6 µg/mL; *p* < 0.05			Not detect	*n* = 105 GDM; *n* = 103 non-GDM	ELISA	[113]
	Lean: 5.0 ± 1.0 µg/mL; Obese non-GDM: 4.1 ± 1.8 µg/mL; Obese GDM: 3.8 ± 1.4 µg/mL; NS	Lean: 20.7 ± 6.2 µg/mL; Obese non-GDM: 20.8 ± 8.6 µg/mL; Obese GDM: 22.0 ± 7.3 µg/mL; NS				Obese GDM *n* = 16; Obese Non-GDM *n* = 13; HP *n* = 15	ELISA	[118]
	IUGR: 4.0 ± 1.6 ng/mL; HP: 3.7 ± 1.1 ng/mL; NS	IUGR: 3.34 ± 1.25 ng/mL; HP: 5.18 ± 3.10 ng/mL; *p* < 0.05				IUGR *n* = 16; HP *n* = 16	ELISA	[115]
	PE: 18.3 ± 2.2 µg/mL; HP: 12.2 ± 1.1 µg/mL; *p* < 0.05				Not detect	PE *n* = 15; HP *n* = 23	RIA; RT-PCR	[77]
	22.6 ± 4.4 ng/mL	8.1 ± 1.1 ng/mL; *p* < 0.05 in relation to maternal plasma				*n* = 27 mothers; *n* = 23 infants	ELISA	[117]
Leptin			Donor: 989 pg/mL; maternal: 1434 pg/mL and 1774 pg/mL after conversion; *p* < 0.05	Donor: 3.8 pg/mL; maternal at the 1st week: 580 pg/mL and 3rd week 577 pg/mL		*n* = 8 infants and their mothers	ELISA	[120]
	Lifestyle interv. 54.2 ng/mL; Standard 54.5 ng/mL; NS	Lifestyle interv. 13.1 ng/mL; Standard 13.1 ng/mL; NS				*n* = 1951 women; *n* = 1174 infants	RIA	[116]
	GDM: ~0.6. ng/mL; non-GDM: ~0.5 ng/mL; *p* < 0.05	GDM: ~0.7 ng/mL; non-GDM: ~0.6 ng/mL; *p* < 0.05			GDM: ~0.5 ng/mg placenta; Non-GDM: ~0.4 ng/mg placenta	GDM *n* = 105; Non-GDM *n* = 103	ELISA	[113]
	FT: 44.5 ng/mL; PT: 75.6 ng/mL; IUGR: 71.8 ng/mL; *p* < 0.05	FT: 19.3 ng/mL; PT: 4.0 ng/mL; IUGR: 1.6 ng/mL; *p* < 0.001		FT 621 pg/mL; PT: 622 pg/mL; IUGR: 844 pg/mL; NS		FT *n* = 16; PT *n* = 16; IUGR *n* = 13	Automated immunoassay analyzer	[119]
	SGA: 19.2 ± 8.4 ng/mL; AGA: 16.7 ± 9.8 ng/mL; LGA: 22.0 ± 11.9 ng/mL; NS	SGA: 3.5 ng/mL; AGA: 6.3 ng/mL; LGA: 9.8 ng/mL; *p* < 0.05				*n* = 56	ELISA	[122]
	PE: 21.7 ng/mL; HP: 4.4 ng/mL; *p* < 0.05	PE: 11.5 ng/mL; HP: 3.3 ng/mL; *p* < 0.05				PE *n* = 45; HP *n* = 45	ELISA	[123]
	EGWG: 14.9 ng/mL; HP 10.4 ng/mL; NS	EGWG: 11.0 ng/mL; HP 7.5 ng/mL; *p* < 0.001				EGWG *n* = 38; HP *n* = 28	ELISA	[121]
	IUGR: 16.0 ± 9.8 ng/mL; Normal: 19.7 ± 16.3 ng/mL; *p* > 0.05	IUGR: 18.0 ± 13.1 ng/mL; Normal: 15.0 ± 8.0 ng/mL; *p* > 0.05				IUGR *n* = 16; HP *n* = 16	ELISA	[115]
	Lean: 9.3 ng/mL; Obese: 46.0 ng/mL; Obese GDM: 42.2 ng/mL; *p* < 0.001	Lean: 5.9 ng/mL; Obese: 7.6 ng/mL; Obese GDM: 6.8 ng/mL; NS				Obese GDM *n* = 16; Obese non-GDM; *n* = 13; HP *n* = 15	ELISA	[118]
	PE: 34.4 ± 3.2 ng/mL; HP: 22.7 ± 2.1 ng/mL; *p* < 0.05				mRNA of leptin in PE > HP	PE *n* = 15; HP *n* = 23	RIA	[77]
Resistin	8.0 ng/mL	17.7 ng/mL				*n* = 109 women and neonates	ELISA	[126]
	10.1 ± 1.1 ng/mL	21.3 ± 1.1 ng/mL; *p* < 0.05 in relation to maternal serum				*n* = 37 women and neonates	ELISA	[125]
	GDM: ~18.0 ng/mL; non-GDM: ~12.0 ng/mL; *p* < 0.05	GDM: ~14.0 ng/mL; non-GDM: ~12.0 ng/mL; *p* < 0.05			GDM 12.0 ng/mg placenta; Non-GDM 9.0 ng/mg placenta	GDM *n* = 105; Non-GDM *n* = 103	ELISA	[113]
	HP: 2.6 ± 0.4 ng/mL; Mild PE: 3.3 ± 0.6 ng/mL; Severe PE: 3.8 ± 0.4 ng/mL; *p* < 0.05				Severe PE > Mild PE > HP	HP *n* = 50; Mild PE *n* = 50; Severe PE *n* = 48	ELISA; Placenta: IHC staining	[124]
	PE: 5.7 ± 0.4 ng/mL; HP: 4.7 ± 0.3 ng/mL; *p* < 0.05				PE ≈ HP	PE *n* = 15; HP *n* = 23	ELISA	[77]
Irisine	662.7 ± 169.5 ng/mL			191 (641.9–30.7) ng/mL		*n* = 81	ELISA	[133]
	151.4 ± 127.0 ng/mL	94.8 ± 77.1 ng/mL; *p* < 0.05 in relation to maternal serum				*n* = 28 pairs mother/newborn	ELISA	[132]
	Non-GDM: 7.7 ± 4.5 pg/mL; GDM: 5.3 ± 0.4 ng/mL; *p* < 0.05	Non-GDM: 5.0 ± 2.1 ng/mL; GDM: 4.9 ± 3.1 ng/mL; NS				GDM *n* = 21; Non-GDM *n* = 21	ELISA	[137]
	28th week GDM 42.1 ± 3.2 pg/mL; Non-GDM 72.9 ± 9.1 pg/mL 6th week postpartum GDM 138.3 ± 6.8 pg/mL; Non-GDM 265.0 ± 40.9 pg/mL; *p* < 0.05			GDM C10.4 ± 4.7 pg/mL; Non-GDM C 57.1 ± 8.3 pg/mL; GDM MM 15.4 ± 0.4 pg/mL; Non-GDM MM 56.4 ± 9.6 pg/mL; *p* < 0.05		GDM *n* = 33; Non-GDM *n* = 22	ELISA	[136]
	GDM: 1679 (13,081–2171) ng/mL; Non-GDM 1880 (1519–2312) ng/mL; *p* < 0.05	GDM: 1723 (1460–1988) ng/mL; Non-GDM: 1257 (1153–1415) ng/mL; *p* > 0.05				GDM *n* = 93; Non-GDM *n* = 97	ELISA	[128]
	GDM: 258.3 ± 127.9 ng/mL; Non-GDM: 393 ± 178.9 ng/mL; *p* < 0.05	GDM: 357.2 ± 248.0; Non-GDM: 333.2 ± 173.4 ng/mL NS				GDM *n* = 20; Non-GDM *n* = 20	ELISA	[131]
	GDM: 482.1 ng/mL; HP: 466.6 ng/mL; POSTPARTUM GDM 446.3 ng/mL; HP 378.0 ng/mL; *p* < 0.05	246.5 ng/mL			53.3 µg/g total protein	GDM *n* = 74; Non-GDM *n* = 74	ELISA	[127]
	Non-lactating: ~400 ng/mL; Lactating GDM at colostral and TM: 250 ng/mL and 360 ng/mL; Lactating Non-GDM at colostral and TM: ~520 ng/mL and 500 ng/mL; *p* < 0.05			C Non-GDM ~550 ng/mL; C GDM ~350 ng/mL; TM Non-GDM ~520 ng/mL; TM GDM ~410 ng/mL; MM Non-GDM ~460 ng/mL; MM GDM ~540 ng/mL; *p* < 0.05		GDM *n* = 15; HP *n* = 15; Non-lactating women *n* = 14	ELISA	[130]
	PT 12.0 ± 2.4 ng/mL; FT 11.5 ± 1.5 ng/mL; *p* > 0.05	PT: 7.7 ± 2.2 ng/mL; FT: 6.8 ± 1.5 ng/mL; *p* > 0.05				PT *n* = 30; FT *n* = 35	ELISA	[135]
			PT 1st week 1.5 μg/mL; 5th week 2.3 μg/mL; FT 1st week 2.2 μg/mL; 5th week 3.5 μg/mL; *p* < 0.05	PT 1st week: 3.2 μg/mL; 5th week: 3.6 μg/mL; FT 1st week: 2.9 μg/mL; 5th week: 3.4 μg/mL; *p* > 0.05, NS		VLBW *n* = 53; FT: *n* = 19	ELISA	[129]
	mild PE VD: during 726.3 ± 102.6 ng/mL, after 834.9 ± 98.0 ng/mL; mild PE CS during 629.9 ± 107.1 ng/mL, after 676.6 ± 99.5 ng/mL; HP VD: during 914.0 ± 90.3 ng/mL, after 975.9 ± 63.7 ng/mL; *p* < 0.05	HP: 97.1 ± 15.7 ng/mL; mild PE VD: 120.9 ± 14.8 ng/mL; mild PE CS: 97.3 ± 16.6 ng/mL; *p* < 0.05				*n* = 150 women and neonates pairs	ELISA	[134]
Ghrelin	HP: 933 pg/mL; EGWG: 1187 pg/mL; NS	HP: 19.5 pg/mL; EGWG: 525 pg/mL; *p* < 0.001				EGWG *n* = 38; HP *n* = 28	ELISA	[121]
	Non-GDM: 439 pg/mL; GDM: 273 pg/mL; T2DM: 239 pg/mL; *p* < 0.001	Non-GDM: 889 pg/mL; GDM: 872 pg/mL; T2DM: 832 pg/mL; NS				GDM *n* = 24; T2DM *n* = 18; Non-GDM *n* = 36	RIA	[139]
		4–5 fmol/mL	15 fmol/mL			VLBW *n* = 25	ELISA	[138]
Nesfatin-1	GDM: 5.5 ± 8.1 ng/mL; Non-GDM: 8.1 ± 23.9 ng/mL; *p* < 0.05	GDM: 5.4 ± 4.0 ng/mL; Non-GDM: 6.2 ± 10.3 ng/mL; NS				GDM *n* = 30; Non-GDM *n* = 30	ELISA	[141]
	non-GDM: 0.9 ± 0.4 ng/mL; GDM: 0.7 ± 0.2 ng/mL; 2nd week of lactation non-GDM: 1.1 ± 0.3 ng/mL; GDM: 0.9 ± 0.2 ng/mL			GDM C: 0.8 ± 0.3 ng/mL; MM: 1.0 ± 0.3 ng/mL; Non-GDM C: 1.6 ± 0.2 ng/mL; MM 1.2 ± 0.4 ng/mL		GDM *n* = 10; Non-GDM *n* = 10	ELISA	[140]
Vaspin	GDM: 0.5 ± 0.2 ng/mL; Non-GDM: 0.8 ± 0.3 ng/mL; 3rd day GDM: 0.4 ± 0.1 ng/mL; Non-GDM: 0.7 ± 0.3 ng/mL; *p* < 0.05				GDM: 0.6 ± 0.3; Non-GDM: 0.7 ± 0.3	GDM *n* = 30; Non-GDM *n* = 27	ELISA RT-qPCR	[142]
Visfatin	2.5–10.7 ng/mL			854–1851 ng/mL		*n* = 24	Not provided	[145]
	18.8 ± 34.3 ng/mL	19.4 ± 4.9 ng/mL				*n* = 20 pairs	Not provided	[144]
	GDM G1: 50.7 ± 14.2 ng/mL; GDM G2: 48.5 ± 14.5 ng/mL; Non-GDM: 31.1 ± 7.5 ng/mL; *p* < 0.05	GDM G1: 35.4 ± 10.4 ng/mL; GDM G2: 36.7 ± 12.0 ng/mL; Non-GDM: 21.0 ± 5.8 ng/mL; *p* < 0.05			GDM G1 52.7 ± 17.3; GDM G2 52.5 ± 9.5; HP 31.8 ± 8.3	GDM *n* = 68; HP *n* = 42	ELISA IHC staining	[143]
	SGA: 1.6 ng/mL; AGA: 4.4 ng/mL; LGA: 1.7 ng/mL; *p* < 0.05	SGA: 5.8 ng/mL; AGA: 3.2 ng/mL; LGA: 5.7 ng/mL; *p* < 0.05				FT *n* = 56	ELISA	[122]
			1st week VLBW ~6.5 ng/mL; FT ~8.5 ng/mL 5th week VLBW 7.5 ng/mL; FT 6.5 ng/mL	approx. 8–10 ng/mL		VLBW *n* = 53; FT *n* = 19	ELISA	[129]
	PE 3.0 ng/m; HP 0.6 ng/mL; *p* < 0.05	PE 3.6 ng/mL; HP 1.0 ng/mL; *p* < 0.05				PE *n* = 45; HP *n* = 45	ELISA	[123]
Chemerin	8–25 ng/mL			C 25–35 ng/mL; TM 20–23 ng/mL; MM 10–20 ng/mL		GDM *n* = 26; Non-GDM *n* = 27	ELISA	[147]
	Lean 125.7 ± 7.4 ng/mL; Overweight 115.1 ± 5.9 ng/mL; Obese 128.3 ± 6.6 ng/mL; Non-GDM 124.2 ± 4.0 ng/mL; GDM 117.6 ± 3.5 ng/mL; NS	Lean 115.0 ± 8.3 ng/mL; Overweight 132.9 ± 8.5 ng/mL; Obese 141.6 ± 7.7 ng/mL; Non-GDM 132.5 ± 4.9 ng/mL; GDM 141.2 ± 4.2 ng/mL; *p* < 0.05			Non-Obese 9 ng/mg protein; Obese 10 ng/mg protein	Non-GDM *n* = 62; GDM *n* = 69; Placenta Non-GDM *n* = 22; GDM *n* = 22	ELISA	[146]
Apelin	Non-obese: approx. 0.3 ng/mL; Obese: approx. 0.3 ng/mL; NS	Non-obese: 0.4 ng/mL; Obese: 0.2 ng/mL; *p* < 0.05			Non-obese 1.0 AU Obese1.3 AU	Non-obese *n* = 36; Obese *n* = 30	ELISA quantitative PCR	[148]
	Non-obese ~0.24 ng/mL; Obese ~0.19 ng/mL; Obese GDM ~0.15 ng/mL; *p* < 0.05			Non-obese: ~12 ng/mL; Obese: ~25 ng/mL; Obese GDM~30 ng/mL		Mother *n* = 13/22; Milk: *n* = 23/25	ELISA	[150]
	GDM: 0.1 ± 0.05 ng/mL; Non-GDM: 0.2 ± 0.09 ng/mL; NS	GDM: 0.1 ± 0.03 ng/mL; Non-GDM: 0.3 ± 0.1 ng/mL; *p* < 0.05				GDM *n* = 24; HP *n* = 21	ELISA	[149]
	GDM: 13.5 ± 8.3 ng/mL; Non-GDM: 9.6 ± 5.9 ng/mL; *p* < 0.05	GDM: 8.8 ± 4.3 ng/mL; Non-GDM: 8.2 ± 1.9 ng/mL; NS				GDM *n* = 30; Non-GDM *n* = 30	ELISA	[141]
Adropin	2.1 ± 0.7 ng/mL			13.7 (30.3–5.2) ng/mL		*n* = 81	ELISA	[133]
	Non-lactating: ~14 ng/mL; Lactating non-GDM: ~16 ng/mL; Lactating GDM: ~7.5 ng/mL; *p* < 0.05			Non-GDM 18 ng/mL; TM 17 ng/mL; MM 15 ng/mL; GDM C 8 ng/mL; TM 13 ng/mL; MM 13 ng/mL		GDM *n* = 15; HP *n* = 15; Non-lactating *n* = 4	ELISA	[130]
	GDM: 2.4 ± 2.0 ng/mL; Non-GDM: 3.3 ± 1.3 ng/mL; *p* < 0.05	GDM: 1.5 ± 0.9 ng/mL; Non-GDM 3.3 ± 1.3 ng/mL; *p* < 0.05				GDM *n* = 20; non-GDM *n* = 20	CLIA	[151]
	PE: 71.2 ± 22.2 ng/L; HP: 100.8 ± 27.0 ng/L; *p* < 0.05	PE: 92.4 ng/L; HP: 106.2 ng/L; *p* < 0.05				PE *n* = 38; HP *n* = 40	ELISA	[152]
	IUGR: 2.6 ± 1.8 ng/mL; HP 5.9 ± 5.9 ng/mL; *p* < 0.05	IUGR: 1.8 ± 0.8 ng/mL; HP 2.8 ± 1.6 ng/mL; *p* < 0.05				IUGR *n* = 16; HP *n* = 16	ELISA	[115]
Copeptin	0.3 ± 0.1 ng/mL			0.6 (2.4–0.4) ng/mL		*n* = 81	ELISA	[133]
	Pregnancy 115.1 ± 41.6 pg/mL; VD 291.7 ± 90.5 pg/mL; CS 215.1 ± 41.1 pg/mL; *p* < 0.05	VD 96.4 ± 18.2 pg/mL; CS 80.8 ± 11.6 pg/mL; NS				*n* = 90 pairs	ELISA	[153]
	10.4 ± 1.7 pmol/L	103.4 ± 22.9 pmol/L				*n* = 66 pairs	FIA	[154]
	Lactating non-GDM: ~8 ng/mL; Lactating GDM: ~8–12 ng/mL; Non-lactating: ~7 ng/mL; NS			Non-GDM C 9 ng/mL; TM 8 ng/mL; MM 9 ng/mL; GDM C 14 ng/mL; TM 11 ng/mL; MM 10 ng/mL		GDM *n* = 15; HP: *n* = 15; *n* = 14 non-lactating	EIA	[130]
	HP: 9.3 ± 0.8 ng/mL; FD: 9.5 ± 2.0 ng/mL; NS	HP: 8.00 ± 4.38 ng/mL; FD: 8.76 ± 1.4 ng/mL; NS				HP *n* = 20; FD *n* = 24	ELISA	[155]
Omentin	26th week GDM 157 ± 83 ng/mL; Non-GDM 158 ± 93 ng/mL 32nd week GDM: 118 ± 77 ng/mL; Non-GDM: 150 ± 89; NS	GDM: 106 ± 61 ng/mL; Non-GDM: 134 ± 45 ng/mL; *p* < 0.05	GDM: 106 ± 61 ng/mL; Non-GDM: 134 ± 45 ng/mL; *p* < 0.05			GDM *n* = 96; Non-GDM *n* = 96	ELISA	[157]
	Non-GDM obese 7.1 ± 0.9 ng/mL; Non-GDM non-obese: 19.5 ± 2.3 ng/mL; GDM obese: 8.2 ± 1.2 ng/mL; GDM non-obese 12.1 ± 1.4 ng/mL; *p* < 0.05	Non-GDM obese 48.3 ± 9.0 ng/mL; Non-GDM non-obese 58.0 ± 6.0 ng/mL; GDM obese 58.3 ± 8.6 ng/mL; GDM non-obese 68.4 ± 8.3 ng/mL; NS			Obese and non-Obese 150 pg/mg protein	Non-GDM *n* = 44; GDM *n* = 39	ELISA RT-PCR	[156]
Dermcidin	~20–25 ng/mL			C70–75 ng/mL; TM 45–55 ng/mL; MM 30–40 ng/mL		Non-GDM *n* = 27; GDM *n* = 28	ELISA	[147]

The table shows adipokine concentration as mean ± SD. Pathological conditions were highlighted in gray. Abbreviations: AGA—Neonates born appropriate for gestational age; C—Colostrum; CLIA—chemiluminescent immunoassay; CS—C-section; EGWG—Neonates born to excessive gestational weight gain; ELISA—Enzyme-linked immunosorbent assay; FD—Fetal disorders; FIA—Fluoroimmunoassay; FT—Neonates born in term; GDM—Gestational Diabetes Mellitus; HP—Healthy Pregnancy; IHC—Immunohistochemistry; Interv.—Intervention; IUGR—Intrauterine Growth Restriction; LGA—Neonates born large for gestational age; MM—Mature milk; NGT—Normal glucose tolerant; non-GDM—non-Gestational Diabetes Pregnancy; PE—Pre-eclamptic Pregnancy; PT—Neonates born preterm; RIA—Radioimmunoassay; RT-PCR—Reverse Transcription Polymerase Chain Reaction; SGA—Neonates born small for gestational age; TM—Transitional milk; T2DM—Type 2 diabetes; VD—Vaginal delivery.

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
