# Peer review of "The Mother–Child Dyad Adipokine Pattern: A Review of Current Knowledge"

_nutrients, 2023, doi:10.3390/nu15184059_

Round 1

Reviewer 1 Report

The mother-child dyad adipokine pattern; A review of current Knowledge

Considerably interesting and well written  long review article focusing upon the mother-child dyad adipokine pattern and the role in fetal development by using 14 adipokines and study their profile in 2 different biological materials maternal plasma and cord.

This review article provides knowledge about the role of the studied adipokines since the majority is identified in two different biological materials, maternal plasma and cord whereas in 4 out of 47 were identified in maternal plasma and milk.                                                                                                                                                                                                                 Table 2. It is better for the reading to have % than numbers of the cases.

Table 1 and 2 must have another column with percentages of the studied adipokines in the sample of the 47 articles selected for the research.

A Table 4 with the % data of the 14 adipokines should be added since it is easier to see the different concentrations

Reference 165 is missing.   

Author Response

Considerably interesting and well written  long review article focusing upon the mother-child dyad adipokine pattern and the role in fetal development by using 14 adipokines and study their profile in 2 different biological materials maternal plasma and cord.

This review article provides knowledge about the role of the studied adipokines since the majority is identified in two different biological materials, maternal plasma and cord whereas in 4 out of 47 were identified in maternal plasma and milk. 

Thank you for the appreciative comment.    

                                                                                                                                                                                  R: Table 2. It is better for the reading to have % than numbers of the cases.

Answer: Table 2 was modified. 

R: Table 1 and 2 must have another column with percentages of the studied adipokines in the sample of the 47 articles selected for the research.

Answer: Table 1 and 2 were modified. 

R: A Table 4 with the % data of the 14 adipokines should be added since it is easier to see the different concentrations

Answer: According to the previous suggestion, we modified Table 1 and 2, the suggested percentage values were added. Additionally in Table 2 details concerning relationships were added. Moreover, the percentage values were added  to the text of the manuscript.

In the light of the above, Table 4 is not necessary. 

R: Reference 165 is missing.  

Answer: Thank you for noticing this.  The reference 165 was instead 164.

Reviewer 2 Report

This manuscript has identified a series of papers that report adipokine concentrations in the mother and offspring during pregnancy and lactation.  It attempts to review relationships between adipokine concentrations from maternal and umbilical blood, milk and placental tissue, where available.

1.  Overall, the presentation of the data could be improved by analysing (1) the normal relationships between adipokine concentrations in the mother and offspring during pregnancy, (2) the effects of gestational age and (3) the effects of pathological conditions separately - rather than by adipokine.  The current organisation of the data is rather dense and difficult to follow in places.  The tables are too large, especially Table 3.

2.  Table 1 could be much more concise to show the number of papers for each adipokine, or even deleted with the information given in the text.

3.  Sorry, I did not understand Table 2.  Maybe more information should be included in the table legend, but it is unclear which relationships are described here.  For example, what does maternal plasma adiponectin have a negative relationship with?  And what does cord plasma adiponectin have a positive relationship with?  I first thought that the relationships were between the maternal and cord plasma samples, but this cannot be the case.

4.  The Abstract could be more informative about the specific findings of the review.  At present, it is rather vague about what the review shows.

5.  The Introduction is rather long and could be much more concise.

6.  What was the specific time frame of article publication?  Lines 154-155 suggest that the articles were published between November 2022 - March 2023, but maybe this was the time of the search?  Were all articles in Pubmed searched up to March 2023, probably not as there would be more than 117?

7.  The Discussion is a little repetitive of both the Introduction and Results.  It should focus more on the implications of the findings, and physiological and pathological significance of these changes and relationships.  What are the known functions of these adipokines during pregnancy that may have long term consequences for the offspring?  To what extent do the concentrations merely reflect body or placental weights or adiposity of the mother and offspring?

8.  There is little in the Discussion about the reasons for variations in concentrations measured between some of the studies (not really due to sample numbers but more likely due to the validity of the assay kits or collection of the samples).

Mostly good, just some of the phrasing is a little awkward, especially in the Introduction.

Author Response

This manuscript has identified a series of papers that report adipokine concentrations in the mother and offspring during pregnancy and lactation.  It attempts to review relationships between adipokine concentrations from maternal and umbilical blood, milk and placental tissue, where available.

  1. Overall, the presentation of the data could be improved by analyzing (1) the normal relationships between adipokine concentrations in the mother and offspring during pregnancy, (2) the effects of gestational age and (3) the effects of pathological conditions separately - rather than by adipokine.  The current organization of the data is rather dense and difficult to follow in places.  The tables are too large, especially Table 3.

Answer: The organization of all Tables was modified. Tables were substantially shortened and additionally Table 3 was reorganized taking into account pathophysiological status. Due to the (1) fact that the presented data are inconsistent, which was highlighted in this work as well as in reviews published recently by Peila and coworkers [2020] and Suwaydi and coworkers [2022] (2) lack of detailed knowledge or huge gaps in many aspects, and (3) finally most of the available studies are restricted to the determination of adipokine patterns, usually only in a single type of biological material from the mother-child system the identification of the real relationships of adipokines pattern and pathophysiological conditions  is  simply impossible or very hindered.

  1. Table 1 could be much more concise to show the number of papers for each adipokine, or even deleted with the information given in the text.

Answer: Table 1 was modified according to the Reviewer’s suggestion.

  1. Sorry, I did not understand Table 2.  Maybe more information should be included in the table legend, but it is unclear which relationships are described here.  For example, what does maternal plasma adiponectin have a negative relationship with?  And what does cord plasma adiponectin have a positive relationship with?  I first thought that the relationships were between the maternal and cord plasma samples, but this cannot be the case.

Answer: According to the Reviewer’s suggestion the additional explanation was included in the Table 2 legend and reported dependencies are presented more clearly. 

  1. The Abstract could be more informative about the specific findings of the review.  At present, it is rather vague about what the review shows.

Answer: The Abstract has been modified and additional sentence was added. Please see page 1, lines: 16. 

“Unfortunately, the available data in that aspect are inconsistent and fragmentary and does not allow for the detailed evaluation of mutual dependencies among analyzed adipokines and risk factors.”

  1. The Introduction is rather long and could be much more concise.

Answer: The Introduction has been modified and shortened.

The sentences from Line 32-34, 38-39, 47-48, 59-62,  and 131-141 were removed.

  1. What was the specific time frame of article publication?  Lines 154-155 suggest that the articles were published between November 2022 - March 2023, but maybe this was the time of the search?  Were all articles in Pubmed searched up to March 2023, probably not as there would be more than 117?

Answer: The sentence in lines 154-155 was modified. Moreover, additional sentences were added. Please see page 4, lines 156-157 and 163-166.

  1. The Discussion is a little repetitive of both the Introduction and Results.  It should focus more on the implications of the findings, and physiological and pathological significance of these changes and relationships.  What are the known functions of these adipokines during pregnancy that may have long term consequences for the offspring?  To what extent do the concentrations merely reflect body or placental weights or adiposity of the mother and offspring?

Answer: The repetitive content from the introduction and discussion has been removed.

Please see Lines: 131-141 and Lines: 830-842. 

  1. There is little in the Discussion about the reasons for variations in concentrations measured between some of the studies (not really due to sample numbers but more likely due to the validity of the assay kits or collection of the samples).

Answer: An additional explanation has been added. Please see Lines: 917-919. 

‘The observed variations in adipokines concentrations between some of the studies are likely the net result of the validation of the assay kits and/or different time points of collection of the biological samples’. 

Round 2

Reviewer 2 Report

The revised manuscript has addressed some of my comments, but it is still very long and could be made much more easy to read.

1.  The 'results' section of the review largely states and repeats the values presented in Table 3.  This is the section that could be made much more concise. 

Also, in Table 3, are these mean +/- SD or SEM values?  And what does '2nd' mean for adiponectin in milk?

  1. In Table 2, the associations are clearer, but what do the variables b, c, d and f mean in a relationship analysis?  Is it the amount/ thickness of preperitoneal fat (measure of adiposity)?  What is maternal health status and the biochemical parameters specifically?  What are the 'subsequent stages'?  This is still difficult to understand in terms of correlation analysis.

  1. Is there nothing to add to the Abstract about the importance of adipokine measurement, and relationships in biological fluids, during pregnancy and lactation, other than that the data are unsatisfactory?

  1. Lines 49-57 could be deleted from the Introduction.  What is 'physiological pregnancy', do you mean 'normal'?

  1. The search period is clearer, thank you, but line 158 states that there were no date restrictions - is this correct if the search was limited to 2000-2023?  And how did you include the criteria requirement for two biological fluids in the search?  The search terms appear to produce tens of thousands of results and it is still unclear how only 117 were identified.  Were PRISA guidelines followed - if so, was this a systematic review?

  1. The Discussion could delete lines 816-828 so that it focuses on the current review.  As mentioned previously: What are the known functions of these adipokines during pregnancy that may have long term consequences for the offspring?  To what extent do the concentrations merely reflect body or placental weights or adiposity of the mother and offspring?

Only minor errors in grammar and phrasing.

Author Response

The revised manuscript has addressed some of my comments, but it is still very long and could be made much more easy to read.

  1. The 'results' section of the review largely states and repeats the values presented in Table This is the section that could be made much more concise. 

A: We thank for this suggestion, however, we consider that tables should be understandable without the need to read the entire manuscript and reversible. 

  1. Also, in Table 3, are these mean +/- SD or SEM values?  And what does '2nd' mean for adiponectin in milk?

A: According to the Reviewer’s suggestion the additional explanation was added to Table 3. Moreover, it was clarified ‘2nd’ in Table 3. 

  1. In Table 2, the associations are clearer, but what do the variables b, c, d and f mean in a relationship analysis?  Is it the amount/ thickness of preperitoneal fat (measure of adiposity)?  What is maternal health status and the biochemical parameters specifically?  What are the 'subsequent stages'?  This is still difficult to understand in terms of correlation analysis.

A: According to the Reviewer’s suggestion the additional explanation was added to Table 2. 

  1. Is there nothing to add to the Abstract about the importance of adipokine measurement, and relationships in biological fluids, during pregnancy and lactation, other than that the data are unsatisfactory?

A: Additional sentences were added to the Abstract. Please see Line: 24-26 and 30-31.

  1. Lines 49-57 could be deleted from the Introduction.  What is 'physiological pregnancy', do you mean 'normal'?

A: The term ‘physiological’ was removed.

  1. The search period is clearer, thank you, but line 158 states that there were no date restrictions - is this correct if the search was limited to 2000-2023?  And how did you include the criteria requirement for two biological fluids in the search?  The search terms appear to produce tens of thousands of results and it is still unclear how only 117 were identified.  Were PRISA guidelines followed - if so, was this a systematic review?

A: The criteria requirement for two biological fluids was obtained by using the word ‘AND’. Based on search strategy and abstract evaluation potential scientific work were relevant. Additional explanation was added to the text. Please see Line: 173. 

For searching we did not use the PRISMA.

  1. The Discussion could delete lines 816-828 so that it focuses on the current review.  As mentioned previously: What are the known functions of these adipokines during pregnancy that may have long term consequences for the offspring?  To what extent do the concentrations merely reflect body or placental weights or adiposity of the mother and offspring?

A: To the Discussion an additional fragment was added. Please see Line: 836-844.